# Intellectual synthesis in mentorship determines success in academic careers

Jean F. Liénard [1,2], Titipat Achakulvisut[3], Daniel E. Acuna[4] & Stephen V. David [1]

As academic careers become more competitive, junior scientists need to understand the value that mentorship brings to their success in academia. Previous research has found that, unsurprisingly, successful mentors tend to train successful students. But what characteristics of this relationship predict success, and how? We analyzed an open-access database of 18,856 researchers who have undergone both graduate and postdoctoral training, compiled across several fields of biomedical science with an emphasis on neuroscience. Our results show that postdoctoral mentors were more instrumental to trainees' success compared to graduate mentors. Trainees' success in academia was also predicted by the degree of intellectual synthesis between their graduate and postdoctoral mentors. Researchers were more likely to succeed if they trained under mentors with disparate expertise and integrated that expertise into their own work. This pattern has held up over at least 40 years, despite fluctuations in the number of students and availability of independent research positions.

[1] Oregon Hearing Research Center, Oregon Health & Science University, Portland, Oregon 97239-3098, USA. [2] Okinawa Institute for Science and Technology, Onna-son, Okinawa 904-0412, Japan. [3] Department of Bioengineering, University of Pennsylvania, Philadelphia, Pennsylvania 19104, USA. [4] School of Information Studies, Syracuse University, Syracuse, NY 13244, USA. Correspondence and requests for materials should be addressed to J.F.Lén. (email: jean.f.lienard@gmail.com)

Most scientific researchers spend several years training under just one or two graduate and/or postdoctoral mentors, suggesting that this small number of relationships can have large impact on their subsequent career. Mentorship is believed to provide both direct intellectual benefits to the trainee—through the learning of new skills and concepts—and indirect social benefits—through engagement with the social network of the mentor[1,2]. Reflecting this widespread sentiment, the stature of mentors and their letters of recommendation are given substantial weight in faculty hiring decisions[3–5]. However, little is known about how the different stages of academic mentorship actually affect the protégé's subsequent career[6,7]. This question is not simply theoretical: identifying the individual determinants of academic success is urgent for trainees searching for faculty positions. More and more postdoctoral fellows are unable to secure a permanent research position even after years of additional training beyond their PhD. Trainees in this position must find ways to extend their postdoctoral training ("permadocs") or join many of their colleagues in dropping out of academic research (the "postdocalypse"[8,9]). Although this issue has gained attention recently, the plight of extended postdoctoral fellowships has been identified since it became a widespread practice, more than 35 years ago[10].

The basic question of what factors lead to success in academic careers is a long-standing one. While early research focused largely on anecdotal studies, the growth of data science has enabled more quantitative approaches to studying this problem[11]. In particular, studies on the science of science have begun analyzing large bibliometric data sets, with the goal of finding the conditions in which scientific breakthroughs are made and published. It has been shown that scientific productivity is highly variable over the course of academic careers[12,13]. The seemingly chaotic trajectories of productivity may especially affect junior scientists, for whom each publication, grant, or collaboration is scrutinized during their competition for positions[14], and who do not have an established scientific reputation to compensate for a gap in publications[15]. Because early career investigators are particularly vulnerable to fluctuations in productivity, the potential benefits of strong mentorship may be particularly valuable at this juncture. A similar quantitative approach can be used to identify the aspects of mentorship that benefit trainees.

Success in academic research careers can be assessed by several different metrics, including publication and citation rates[16], funding levels[17], and a protégé's own mentoring achievements. Academic proliferation (the number of progeny trained by a mentor, sometimes termed academic fecundity) provides a measure of this last metric[18,19]. Empirical studies have found the number of academic progeny to be correlated with academic achievements, such as holding a position at a prestigious institution, holding a named chair[20], publishing more papers[19], or receiving the prestigious Nobel prize[21]. Thus academic proliferation provides a proxy for these other measures of success. Academic proliferation gives insight into two aspects of research careers: (1) attrition rate, where a researcher who has never mentored someone else probably does not hold a permanent position, and (2) scientific proficiency, where more successful mentors have a greater number of trainees. This second effect might reflect that greater fame attracts more students, greater financial resources allows more hires, and a virtuous circle where trainees contribute back to the prestige of the mentor through collaboration and contribution to an extended social network throughout their own careers[2].

Given the central role that mentorship plays in academic research, studying a large network of mentors and trainees has the potential to provide insight into the drivers of academic success. The Academic Family Tree (academictree.org) is an online effort begun in January 2005 to document training relationships in a relational database. This project originally started with a focus on the field of neuroscience but progressively expanded to span more than 50 disciplines[18]. Researchers in the database are linked to publications they have authored by an automated record linkage to the Medline and Scopus databases. In the current study, we applied a data-driven approach to study 500,000+ life science researchers, predominantly from neuroscience, with a focus on the subset with documented graduate and postdoctoral training. Our objective was to uncover how patterns in the network of mentors and protégés shape their academic success: to what extent does mentorship impact the future career of trainees? What is the relative influence of social versus intellectual factors on mentoring relationships? Do graduate or postdoctoral mentors have a greater impact on trainee careers? What are the long-term temporal trends that influence the success of trainees?

To address these questions, we measured several properties of the mentor network graph and of semantic relationships between publications by mentors and trainees. We then used a regression framework to quantify the impact of these different factors on the two outcome variables defined above: acquiring an independent research position after postdoctoral training and the academic proliferation of those who do obtain independent positions. Our analysis revealed that several factors had significant predictive power for the success of independent careers. Trainees of graduate and postdoctoral mentors with high proliferation tended to be more successful themselves, consistent with the previous observations for graduate mentors[19]. In addition, success rates were predicted by the pattern of intellectual similarity between mentors and trainees. Trainees whose research was able to synthesize the influence of mentors with distinct expertise had larger odds of continuing in research. Thus a model emerges in which the most successful trainees are trained by successful mentors and are able to synthesize content of both mentors' work in their own research.

## Results

**Properties of mentorship networks**. To study the influence of graduate versus postdoctoral mentorship on trainee success, we focused our study on "triplets" of researchers, each consisting of a trainee, a graduate mentor, and a postdoctoral mentor. These data were drawn from all life science fields represented in the Academic Family Tree. We evaluated several population-level features of the triplets, which fall broadly into two categories: graphical properties of the mentorship network and semantic properties of mentor and trainee publications (Fig. 1). The first group of properties includes the year in which training was completed, duration of postdoctoral training, proliferation rate of mentors and trainee (average number of trainees per decade), professional age of mentors (years since the mentor completed their own training), and mentor network distance (the distance to the mentors' earliest common ancestor, Fig. 1a). The second group includes the similarity of scientific output by each pair of researchers in the triplet, measured by latent semantic analysis of published abstracts (Fig. 1b, c)[22]. The specifications of these variables are detailed in the Methods.

The primary measure used to evaluate success of academic careers of both trainees and mentors here is academic proliferation rate, defined as the average number of researchers trained per decade. Across all triplets, both trainees and their mentors show overall similar distributions of proliferation rates, with a mode of about one (i.e., one trainee every 10 years, Fig. 2a). Mentor proliferation rates are well-described by a Poisson distribution. However, 60% of the protégés did not train anyone themselves,

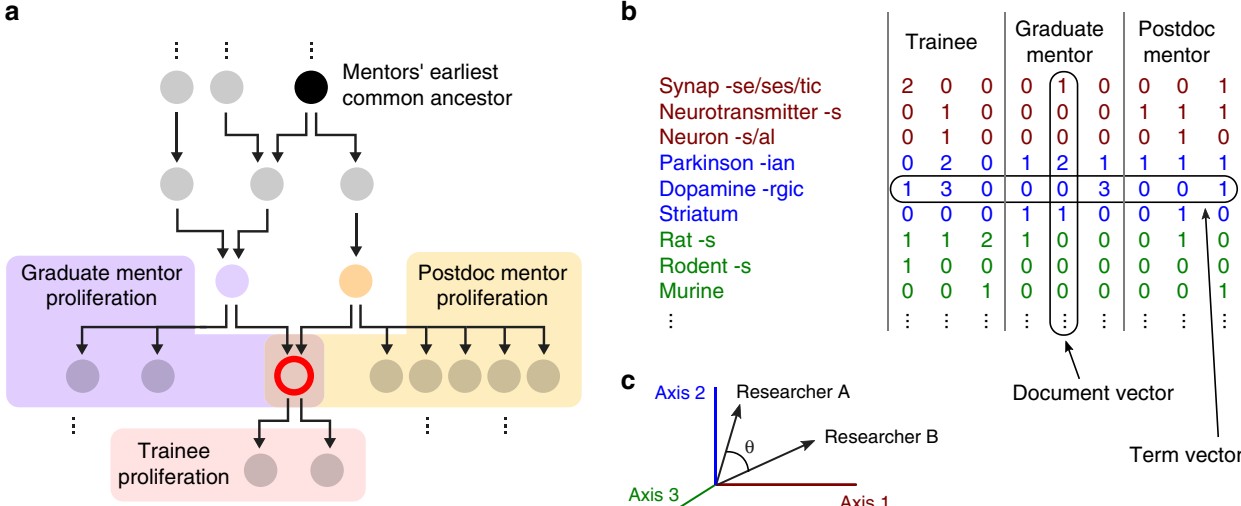

**Fig. 1** Predictor variables. **a** Schematic mentorship network graph for a single trainee (red), with mentor–trainee relationships indicated by arrows. Analysis focused on researchers who completed training with at least one graduate (purple) and one postdoctoral mentor (yellow), the three together comprising a mentorship triplet. Mentor network distance is computed as the distance to their earliest common ancestor (black). Professional success is measured by the proliferation rate, the average number of trainees per decade after the start of a researcher's independent career. **b** Intellectual relationships were characterized by semantic analysis of abstracts indexed in the Medline and Scopus publication databases. Semantic content was quantified by a term-frequency—inverse document frequency (TF-IDF) metric, in which stemmed words were counted and their relative frequency was used to define a space for principal component analysis. **c** The term vector for each publication abstract was projected onto a vector spanning the 400 largest principal components. Publication similarity was computed as the cosine distance between the average publication vector for each researcher, after excluding their commonly authored publications

reflected by a large peak at zero proliferation rate. Their distribution is better described by the product of binomial and Poisson distributions (see below and Eq. 1). The proliferation rates of graduate and postdoc co-mentors within a triplet have low correlation (Pearson's coefficient $r = 0.14$ with 95% CI [0.12, 0.16]), producing a wide distribution of the difference in proliferation rates (Fig. 2b, blue bars). This distribution is not symmetric. Postdoctoral mentors had significantly greater average proliferation than graduate mentors (mean difference: 4.50, 95% CI [4.24, 4.75]). Although their proliferation rates differed, co-mentors tended to be closer in the mentorship network graph than expected by chance (Fig. 2c). Protégés in triplets were trained by mentors with an average academic age of 10 years (Fig. 2d). Graduate mentors tended to have slightly lower academic age than postdoc mentors, possibly reflecting budget constraints on hiring more costly postdocs during the early independent career. This age difference is stable through time.

Some long-term trends are readily visible in the dataset. Most triplets in the analyzed data completed training after 1990 (Fig. 2e). This upward trend reflects the ongoing growth of postdocs in life science[23], the number of which increased by a factor of four over the period 1980–2010[6]. The 10-fold increase reported in Fig. 2e is larger than this trend. This difference may reflect the substantial recent growth of neuroscience[24], which is well-represented in our dataset. It may also reflect a sampling bias in the database favoring more recent graduate students and postdocs. In parallel with the growing number of postdocs, the data also indicate an increase in the duration of training over time (Fig. 2f). This increase is true for the duration of both graduate and postdoctoral training, the latter of which has increased by an average of about a year since the 1970s.

Research performed as a graduate student or postdoc may be more or less aligned to the mentor's own research. To study how the similarity of intellectual output between mentors and their trainees impacts the subsequent success of trainees, we performed a latent semantic analysis on the abstracts of non-coauthored

papers published before the end of postdoctoral training (see Methods and Fig. 1). Not surprisingly, co-mentors tended to have more similar semantic content in their publications than randomly selected pairs of researchers (Fig. 2g). Their publication similarity increased with proximity in the training network, as measured by mentor graph distance (Fig. 2h). However, graph distance is not the only factor influencing the publication similarity. Co-mentors displayed a greater similarity than randomly chosen pairs of researchers with the same graph distance (Fig. 2h). Thus, the relatively high publication similarity of co-mentors reflects factors beyond the similarity of their academic genealogy.

**Intellectual synthesis and continuation in academic research.** Academic research careers adopt many different shapes and may involve a mix of research and university-level teaching. Here, we focus specifically on a criterion that provides a proxy for success of academic research careers in life science, and which is accessible in our dataset: the training of at least one graduate student or postdoctoral fellow. Indeed, the training of a junior researcher is a years-long commitment, and a stable research position is often an institutional prerequisite for it. Conversely, virtually all successful researchers in life science manage a team composed of graduate students and postdocs.

Using publication similarity as a measure of intellectual overlap between researchers, we considered its relationship to the odds of becoming a mentor, that is, for the trainee to continue in academia and themselves train at least graduate student or postdoctoral fellow. We observed several significant correlations between publication similarity and trainee success. In particular, greater publication similarity between a trainee and each mentor led to higher probability of continuing in academia (Fig. 3a, b, $p < 0.001$, two-sample Kolmogorov–Smirnov test). In contrast, co-mentors with greater publication similarity had trainees less likely to continue in research (Fig. 3c, $p < 0.001$). There is no obvious

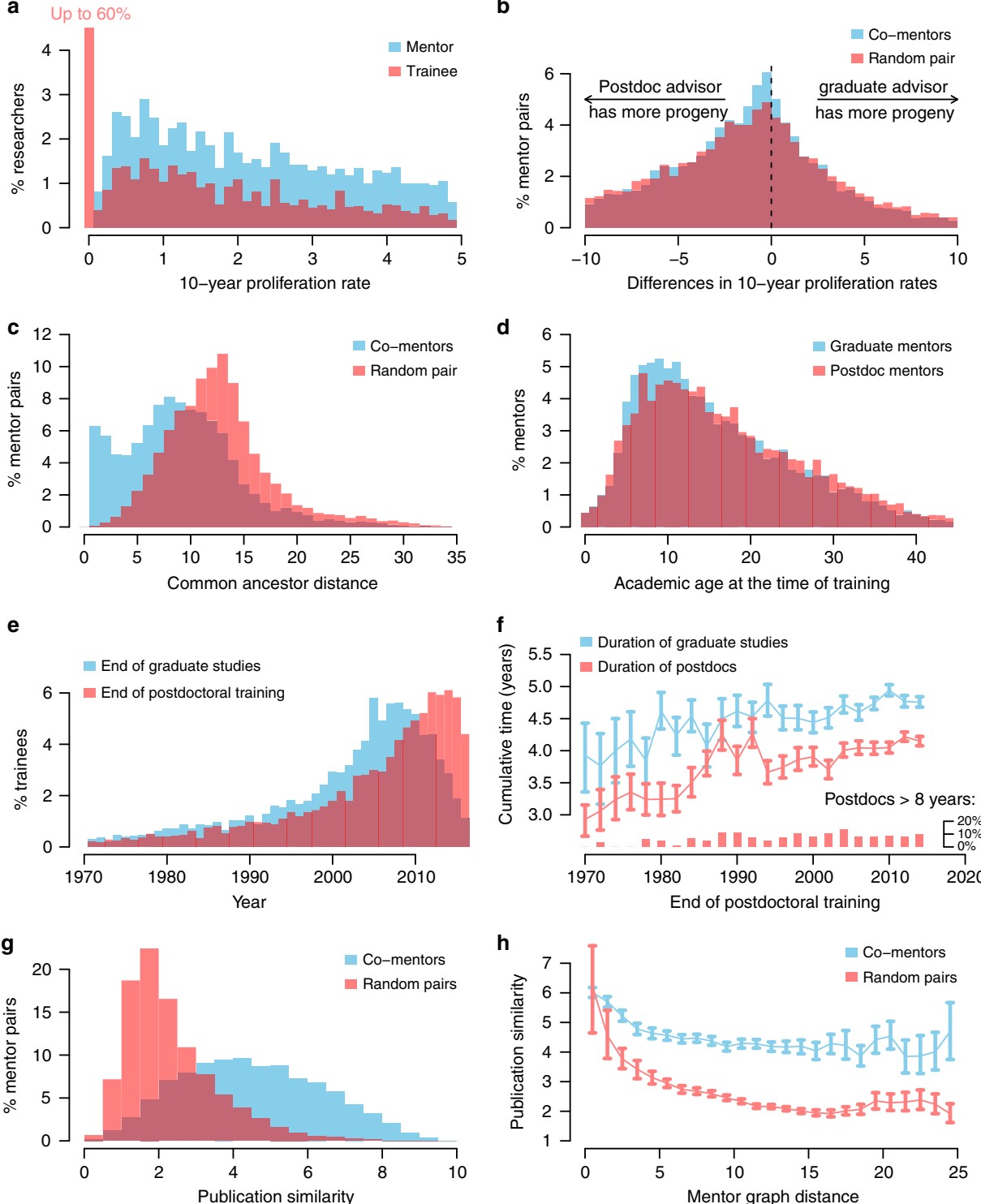

**Fig. 2** Main features of the $n = 18,856$ mentorship triplets in life science analyzed in this study. Each triplet is constituted by a pair of graduate and postdoc mentors and their common trainee. **a** Distribution of proliferation rate (average number of trainees per decade) of mentors (blue) and trainees (red). The large peak at zero for trainees reflects the large number of trainees that do not go on to mentor students of their own. **b** Difference in proliferation rate between graduate and postdoc mentors common to triplets (blue) and mentor pairs picked at random (red). **c** Common ancestor distance of co-mentors (blue) and mentors picked at random (red). **d** Academic age at the time of training for graduate and postdoc mentors of each triplet. **e** Year of graduation (blue) and end of postdoc (red) for triplets. The decreasing number of trainees graduated in recent years, in blue, results from the inclusion criterion of trainees that were selected as those having undergone postdoctoral training. **f** Mean cumulative time spent in graduate studies (blue) and postdoctoral fellowships (red), as a function of postdoc end date. Postdoctoral studies lasting more than 8 years were excluded from the computation of cumulative time. Instead their proportion was plotted as bars at the bottom the graph. **g** The similarity between co-mentors (blue) is higher than among a randomly picked pair of researchers (red). **h** Closer common ancestor distance leads to greater publication similarity, and this effect is cumulative with the higher proximity of researcher that co-mentor the same trainee

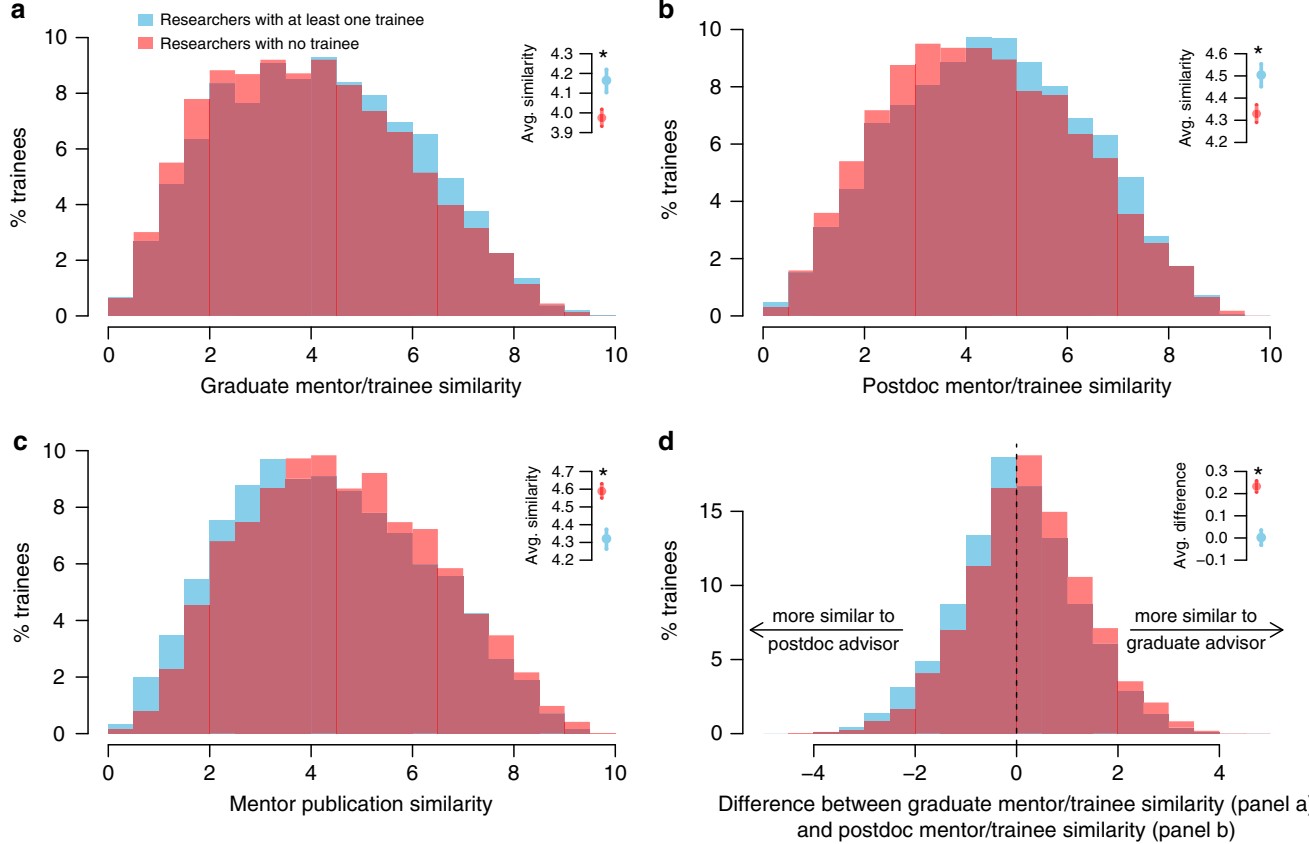

**Fig. 3** The odds of becoming an academic mentor are correlated with a trainee's ability to synthesize disparate influences from their own mentors, as measured by the semantic similarity of their abstracts published prior to the end of postdoc training. Insets indicate the mean of the two trainee groups, i.e., those who do and do not become academic mentors, and 95% confidence intervals. **a, b** Trainees who become mentors show greater similarity with their graduate (A) and postdoctoral (B) advisors. **c** Lower similarity between these mentors is also linked with better odds to continue in academic research. **d** protégés that have a greater publication proximity with their postdoc mentor, compared to their graduate mentor, tend to move to independent academic positions more frequently

connection between the differences in publication similarity in Fig. 3a–d and the proliferation rate of mentors, ruling out that these effects are linked to the mentor's training track-record (Supplementary Fig. 1).

Together, these observations are consistent with a model in which a trainee who successfully synthesizes knowledge and approaches from dissimilar mentors is more likely to continue on to an independent academic career. Furthermore, successful trainees tended to show closer semantic proximity to the postdoctoral mentor than the graduate mentor (Fig. 3d, *p* < 0.001), suggesting that the postdoctoral relationship is a stronger determinant for the trainee's future employment.

**Model of academic success in life science.** The patterns in Figs. 2 and 3 suggest a link between the characteristics of mentors and their trainee's odds of staying in academia. At the same time, the relatively strong coupling between variables, such as postdoc duration and training end date (Fig. 2f) and mentor graph distance and publication similarity (Fig. 2h), presents a challenge for building the predictive model of trainee success. In order to disentangle these factors, we developed a statistical model to measure the impact of each variable on the subsequent career of trainees. Our model considers two possible scenarios (Eq. 1): (a) the protégé moves on to a private sector position or to a public research position that does not involve training, thus excluding them from having descendants in the training network; and (b) the protégé moves on to an independent research position

involving training, in which case their own proliferation rate is used to measure their human-capitalized scientific legacy. In order to permit time for measuring trainee proliferation over a 10-year window, we restricted our dataset to triplets where the protégé finished their latest training no later than 2007. All the variables were available at the completion of postdoctoral training and thus were not biased by the subsequent independent career of protégés.

These scenarios were integrated into a mixed model that predicted the proliferation rate of trainees based on variables characterizing their mentor relationships. We simultaneously determined the best model architecture (hurdle or zero-inflated with Poisson or negative binomial distribution, see Methods) and the best combination of variables through a cross-validated search conducted over all predictor combinations. The best-fitting model overall was the zero-inflated negative binomial model (Supplementary Table. 2). We focus on this model for the remainder of the paper. Quantitative details of the model comparison, including the cross-validated log-likelihood scores of alternative architectures, are available in Supplementary Information.

The relevance of individual variables was assessed using their Shapley scores[25], which indicate their relative contribution to the overall goodness-of-fit (Table 1), and using their ranks from the Forward and Backward Selection Algorithms (FSA and BVA, see Methods). Several variables had a negative Shapley score, indicating that they tend to form spurious relationships with the protégé's proliferation. In particular, the mentor network distance, academic age of graduate mentor, and postdoctoral

**Table 1 Overview of factors impacting trainee proliferation and their contribution to the model's goodness-of-fit**

| | Goodness-of-fit | Variable ranks | | |
|---|---|---|---|---|
| | | SV | FSA | BSA |
| *Temporal trend* | | | | |
| **Training end year** | 1.822 | 1 | 1 | 2 |
| Postdoc duration | −0.027 | 10 | 11 | 11 |
| *Network* | | | | |
| **Graduate mentor proliferation** | 1.301 | 3 | 3 | 3 |
| **Postdoc mentor proliferation** | 1.764 | 2 | 2 | 1 |
| Mentor graph distance | −0.109 | 12 | 12 | 12 |
| Graduate mentor age | −0.010 | 8 | 7 | 8 |
| **Postdoc mentor age** | 0.121 | 6 | 6 | 7 |
| *Publication* | | | | |
| **Mentor publication similarity** | 0.369 | 4 | 4 | 5 |
| Graduate mentor/trainee similarity | −0.025 | 9 | 10 | 10 |
| **Postdoc mentor/trainee similarity** | 0.163 | 5 | 5 | 4 |
| **Publications with graduate mentor** | 0.090 | 11 | 9 | 6 |
| **Publications with postdoc mentor** | −0.004 | 7 | 8 | 9 |

Variables in bold were kept in the model. The contribution of each variable to the overall model performance is its Shapley value, with positive values denoting a greater contribution. The 12 variables are ranked according to their increasing order of importance, according to their Shapley Value ("SV"), the Forward Selection Algorithm ("FSA"), and the Backward Selection Algorithm ("BSA")

training duration had negative Shapley scores. These same variables were also excluded by the iterative variable selection (both FSA and BSA, Table 1). Thus, we excluded them from further analysis. The relevance of the number of publications with the postdoc mentor was more ambiguous: it had a negative Shapley value but was ranked above the number of publications with graduate mentor by the forward CSA algorithm. Also, this metric is widely used to evaluate job applicants and has been reported as the most important metric used by search committees, above the quality of journals or the funding track-record[26]. Thus we opted to include it in subsequent analyses.

**Determinants of academic success**. The impact of mentorship variables on the odds of trainees obtaining an independent research position and on their long-term proliferation rates are summarized in Table 1 and Fig. 4. The odds of continuing in academia were positively influenced by higher mentor proliferation rates, greater postdoctoral mentor academic age, and close publication similarity between the protégé and their postdoctoral mentor (Fig. 4 and Supplementary Fig. 12B and C). In contrast, training end year and high mentor publication similarity negatively influenced the probability of continuing in academia (Fig. 4 and Supplementary Fig. 12A and D).

A similar, but not entirely overlapping set of variables influenced the protégé's long-term proliferation rate. Trainee proliferation was positively influenced by the mentors' proliferation rates, along with semantic proximity to the postdoctoral mentor and the number of publications co-authored with the graduate mentor (Fig. 4 and Supplementary Fig. 12G–I). Overall, these results show that highly prolific mentors tend to provide their protégés with the assets required for their own success in academia, both in terms of securing permanent research positions and of increasing their long-term proliferation.

The effect of training end date on the odds of continuing in academia was found to be very strong (Fig. 4 and Supplementary Fig. 12A), consistent with known long-term trends toward a decreasing number of independent academic positions available to postdoctoral trainees (e.g., refs. 10,27,28). We considered the possibility that temporal bias in the sampling of other variables could confound their effects with this strong temporal trend. To control for this possibility, we fit the same models using a temporal subset of data and without the training end date

(Supplementary Fig. 5). We also evaluated alternative temporal models, which included either a polynomial expansion of training end date or training end date as an ordinal variable coding for different temporal epochs (Supplementary Figs. 6 and 7). These control models reveal the same effects for the non-temporal variables, confirming that the significant effects in Fig. 4 are not confounded by temporal trends.

Characteristics of the postdoctoral mentor generally had greater influence on trainee success than the graduate mentor. This suggests a dominant role of postdoctoral mentors on the future career of protégés. The age of the postdoctoral mentor contributed to the likelihood of the protégé securing a permanent position, with increased odds for older postdoctoral mentors. In contrast, graduate mentor age did not have a significant influence. We tested whether a possible influence of the graduate mentor's seniority was masked by correlations with features of the postdoctoral mentor. We fit an alternative, partial model based only on features of the graduate training. This model ruled out the possibility of confounds with the postdoctoral mentor, as it produced coefficients similar to the full model, again revealing no influence of graduate mentor academic age (Supplementary Fig. 2).

Network variables were broadly found to make a larger contribution to overall model performance than publication variables (Fig. 5). This relatively greater influence is consistent with the Shapley values and the variable ordering in Table 1, and is found across all formulations of the model studied (Supplementary Table 2).

We restricted our main analysis to data that was available before the end of training to avoid any confound associated with continuing versus not continuing in academia. To investigate whether the semantic content of papers published after the end of the postdoc continue to influence the career outcomes, we further included them as extra variables in the model. We observed substantial explanatory power for this late-publication similarity in explaining continuation in academia, and specially so for the postdoc advisor–trainee similarity (Supplementary Fig. 13). This finding suggests that strong ties formed during training and transitioning into a collaboration with the former advisors has a beneficial impact on the trainee's career. This also reinforces the idea that the postdoctoral advisor has a larger influence on the future career than the graduate advisor (Fig. 4), as was found

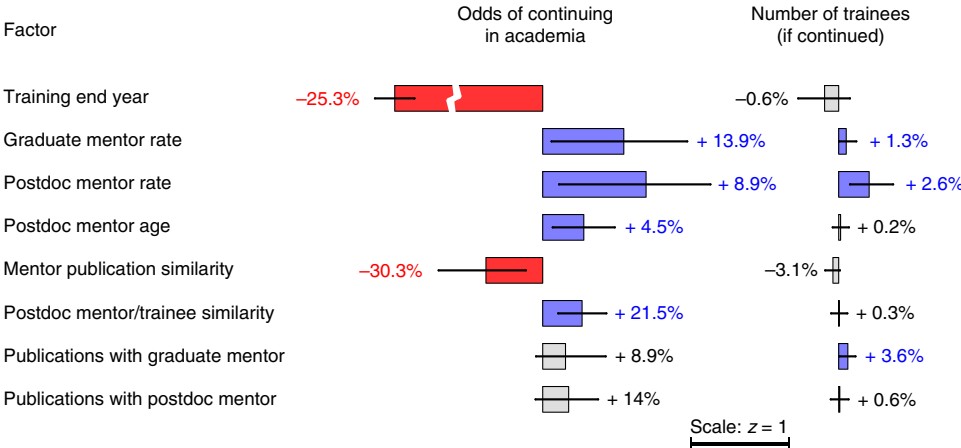

**Fig. 4** Modeled effects of variables for continuing in academia (left column) and for training rate when continuing (right column). Percent values indicate the change in the odds of continuing in academia (left) and in the training rate (right) following an increase of one unit for each factor. Error bars show the 95% bootstrapped confidence interval for the modeled effects. The bars and error bars are plotted in z-scored units (cf. scale bar), enabling comparison of their effects despite their different scales. Significant changes are color-coded and the associated percentages are shown in bold. For example, the interpretation of "Postdoc mentor rate" is as follows: ceteris paribus, an increase of 1-point on the postdoc mentor proliferation scale (i.e., one more trainee per decade) improves the odds of the protégé to find a permanent position by 8.9%, and this effect is statistically significant. The long-term effect of this change is also significant, and the protégé's proliferation rate is then increased by 2.6%

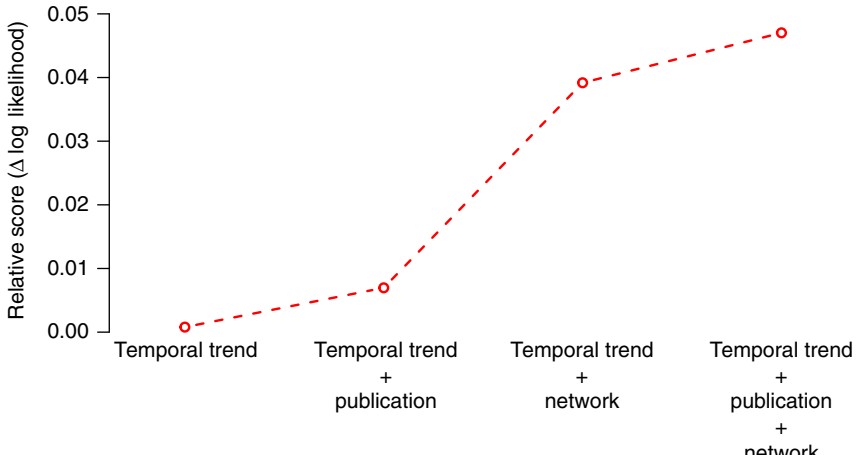

**Fig. 5** Impact of adding the different groups of variables on the log-likelihood of the model. Once the temporal trends are accounted for, the largest effect corresponds to the inclusion of mentor network features (training rate and academic age)

using variables available at the end of the postdoc (Supplementary Fig. 13).

**Consistency of effects across fields**. The composition of the life science dataset is dominated by neuroscience graduates. Indeed, 62% of the triplets ($n = 14{,}953$) have a trainee identified as belonging to the field of neuroscience, and the remaining 38% ($n = 5742$) span several other fields. To assess the consistency of the effects across fields, we split the data into two subsets: neuroscience only and other life sciences. These nonoverlapping datasets show similar, albeit noisier, patterns compared with the full dataset reported in main text (Fig. 2 vs. Supplementary Figs. 8, 9). The mentorship patterns of Fig. 2a–d are comparable across subsets. Both also show the same trends of increasing postdoctoral trainee numbers and training duration (Fig. 2e, f) and the same patterns of publication similarity (Fig. 2g, h). Importantly, models computed for both data subsets showed the positive effect of intellectual synthesis, with a strong effect of mentor publication similarity in both cases (Supplementary

Figs. 10 and 11). Thus, the advantage of trainees performing intellectual synthesis generalizes across the life sciences, although the significance of some variables is not achieved in the smaller sample of the non-neuroscience dataset. We also remark that the proliferation rate of the graduate mentor shows a substantially stronger influence in the non-neuroscience subset, along with the effect of postdoctoral publication similarity. Whether this reflects a differential impact of the postdoctoral advisor between neuroscience and other life sciences, or whether this is an artifact of limited sampling remains to be investigated.

**Nonlinear influence of mentor graph distance**. Mentor graph distance showed a clear inverse relationship with mentor publication similarity (Fig. 2h), which has large explanatory power in the model (Table 1). Indeed, these two variables have a weak but significant correlation ($r = -0.192$ and $p < 0.05$, Fig. 6a). Thus, although mentor graph distance had low Shapley value and low importance according to the forward and backward CSA algorithms (Table 1, Fig. 6b), we considered the possibility that it

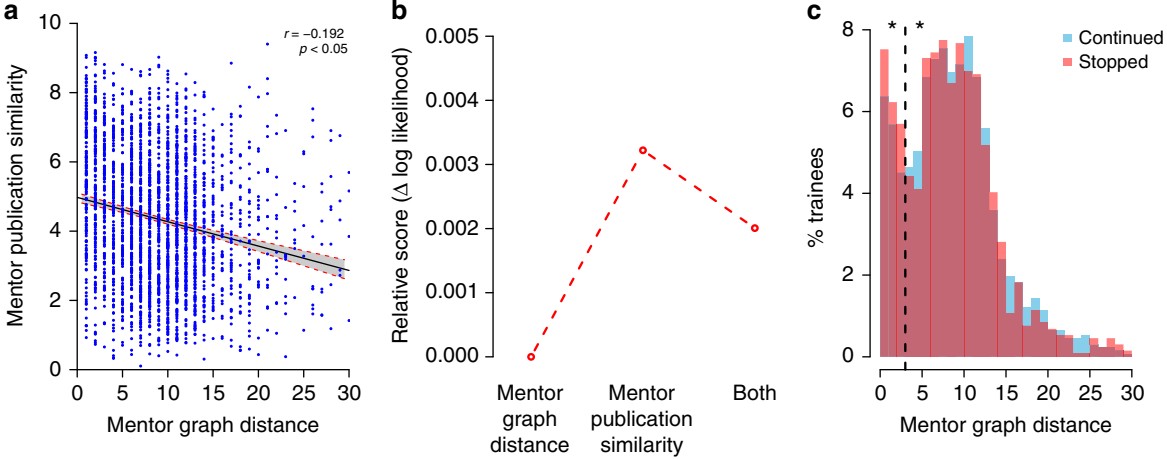

**Fig. 6** Relationship between the mentor early similarity in publications and their common ancestor distance, and their respective predictive power in the model. **a** Scatter plot with linear regression, the shaded area displays the 95% confidence interval in prediction. **b** Relative contribution to the model goodness-of-fit, when acting in isolation or together. **c** Distribution of common ancestor distance for the trainees who stopped (red) or continued (blue) an academic research career. The dashed line indicates the boundary between the two peaks in the distribution of mentor graph distance. Probability of continuing is significantly smaller for the group with shorter graph distance

might somehow influence trainee outcomes. Mentor graph distribution shows a striking bimodal distribution that suggests a more complex nonlinear relationship with other model variables (Fig. 6c). The distribution of mentor graph distance is broadly similar for trainees who did or did not continue in academia. However, for trainees with very short mentor graph distance (< 4 steps) the probability of continuing in academia appears to be consistently lower. We grouped the data into two categories, tight-knit mentorships (with mentor graph distance less than four), and out-of-nest mentorships (with mentor graph greater than or equal to four). In this case, we do see a different distribution of tight-knit and out-of-nest mentorship groups for the two different outcomes ($p = 0.0072$, Pearson's $\chi^2$ test using 100,000 Monte Carlo permutations). Thus trainees of advisors that are closely connected in the mentoring graph may be at a disadvantage in acquiring independent research positions, although a larger dataset will confirm that this effect is a confound with other features, in particular the high publication similarity associated with closely related mentors.

**Interactions between network and semantic variables**. Thus far, we have identified factors correlating with academic success, including a statistically more important role of postdoctoral advisors on trainee careers relative to graduate advisors. The interpretation that postdoctoral advisors contribute more to trainee success is consistent with them playing a critical role as postdocs build up their CV and professional skills to gain an edge in the competition for permanent positions.

However, this causal interpretation of postdoctoral mentor influence could be confounded by trends guiding the trainee's selection of a postdoctoral advisor. In this case, the positive features of postdoctoral advisors would merely be side effects of a trainee strategy in selecting their mentor. A specific possibility is that there is a systematic benefit associated with trainees following a trajectory of "upward mobility", moving from a less prestigious graduate mentor to more prestigious postdoctoral mentor. In the framework of the variables investigated here, this effect would correspond to trainees moving to a postdoctoral mentor with a greater proliferation rate than the graduate mentor. To test this hypothesis, we re-fit the model with a new interaction term, computed as the ratio of postdoctoral/graduate mentor

proliferation rates. This additional term held no significant predictive power (cf. Fig. 3 in Supplementary Information), leading us to infer that the strategy of moving to a more prestigious postdoctoral mentor is not the common pattern for successful trainees.

Alternatively, it may be that trainees who move to a more thriving subfield of research for postdoctoral training have more chances to get a permanent position. In this other scenario, what really matters is the scientific mobility of trainees, and not the features of their postdoctoral advisors per se. We evaluated this possibility by re-fitting the model with two additional interaction terms, computed as the product of the trainee–postdoctoral advisor similarity and the postdoctoral advisor proliferation rate ("Postdoc mentor rate × similarity"), balanced with the same metric computed for the graduate studies ("Graduate mentor rate × similarity"). By design, the former term should be high for trainees who shift their research focus to the "better" field of their postdoctoral mentor, while the latter should be high for trainees who stayed in the "better" subfield of their graduate studies. Again, we fail to see a significant effect of either of these interactions on the odds of finding a permanent position (Fig. 4).

## Discussion

We identified factors related to mentorship that influence the academic success of postdoctoral trainees in biomedical research. We considered two measures of success: whether or not a trainee obtains an independent research position and their proliferation rate (number of researchers trained) once a position is obtained. The factors influencing the likelihood of trainees obtaining an independent position were (a) the year of entry to the job market (reflecting long-term trends in job openings), (b) the success of their mentors (i.e., mentor proliferation rate), (c) their ability to synthesize between the intellectual output of their mentors, and (d) the professional age (time since graduation) of the postdoctoral mentor. The main predictors of trainee proliferation were the mentors' training rates and publishing research that was similar to that of their graduate mentor. Neither the duration of training nor the professional age of the graduate mentor impacted trainee success. Below, we discuss these findings and their relevance to the existing literature.

This study reveals the importance of intellectual synthesis in the pursuit of an independent research career in life sciences. Trainees tended to be more successful if publications by their graduate and postdoctoral mentors had low semantic similarity, suggesting that their work links ideas and/or approaches from two previously disparate subfields. This finding can be framed within the weak-ties theory of Granovetter[29], which emphasizes the importance of bridges between weakly connected communities. Trainees with disparate mentors may also benefit from more professional opportunities in the larger combined network of their advisors[29,30]. In addition, researchers bridging disparate scientific communities are in a position to diffuse their innovations across a larger group of peers, possibly garnering more recognition for their work[31].

For these beneficial effects to occur, the trainee's research must actually bridge the disparate training fields. Thus, it is important that the work maintains some similarity with that of both the graduate and postdoctoral mentors. Indeed, successful trainees tended to have strong semantic similarity to both mentors, consistent with a meaningful intellectual impact by both on the trainee's work. This effect persists and is amplified when considering publications after the completion of postdoctoral training, suggesting a long-lasting benefit of synthesis.

Given that trainees benefit from mentors with dissimilar research, one might also expect trainees to benefit from mentors separated by a large distance in the genealogy graph. However, mentor network distance (i.e., distance to a common ancestor) does not predict trainee continuation or proliferation (Table 1, Fig. 6b). This suggests that mentor graph distance may be too crude a measure of intellectual similarity to provide significant predictive power. Alternatively, different social factors may influence the career path of trainees with closely related mentors. The bimodal distribution of mentor graph distance suggests that there are in fact two groups of trainees. The decreased odds of having scientific progeny for the group with mentor network distance less than four is consistent with the hypothesis that a lack of intellectual or social diversity is detrimental to professional success.

We also found that mentors' proliferation rates positively influence the trainee's proliferation rate, consistent with previous findings[19–21]. The link between mentor and trainee success in academia has been observed using other measures. In particular, mentor prestige has been shown to be correlated with trainee publication rate[32], and a mentor's research productivity impacts both the prestige of a trainee's first professional research position and research productivity later in their career[33].

For our bioscience dataset, we also found that the proliferation of the postdoctoral mentor has a greater effect than that of the graduate mentor on the odds of securing a permanent position (Fig. 4). This observation is consistent with Long et al.[3], which showed that the prestige of the postdoctoral institution has a stronger positive effect than that of the doctoral institution for a cohort of biochemists graduated between 1957 and 1963. The greater influence of the postdoctoral advisor is also consistent with the idea that the postdoc is a launching pad for an independent career, during which time scientists build up critical secondary skills (network connections, publication and grant funding track-record, technical expertise) required for an independent position. Our analysis also suggests that the benefits of training with a successful mentor and pursuing research along similar intellectual lines may be related, as successful mentors are likely to work on topics of broad relevance to their field. Overall, our study supports the long-standing advice that prospective students should look at the training record of potential mentors to assess their quality[34,35].

It is sometimes considered that the duration of postdoctoral training has increased in recent decades. Quantitative reports have showed a stabilization (e.g.,[28]). However, most previous work draws on data from the Survey of Doctorate Recipients (SDR), which is limited to US-graduated researchers. Thus it omits data for international postdocs, who often remain longer in postdoctoral positions due to visa limitations[36]. In the current study, we find that the mean duration of postdoctoral training in life sciences has increased, from less than 3 years in 1970 to 4.1 years in 2015 (Fig. 2f). The proportion of long-term postdocs (> 8 years) has remained stable at around 10%. The absence of an increasing trend in long-term positions may be seen as encouraging. However, the persistently large number of "permadocs" remains worrisome and fits in the narrative that sometimes postdocs are more a source of cheap labor than a meaningful step on the pathway of career development[37].

Previous work on the influence of postdoctoral training duration on the odds of securing a permanent position show inconsistent results. Yang and Webber[38] recently showed that completing two to four postdoctoral appointments nearly doubled the odds of obtaining an academic position, although it did not enhance long-term productivity. In a study on biochemists, Nerad and Cerny[27] showed, on the contrary, that relatively short postdocs (< 5 years) led to better outcomes in academic careers. The beneficial experience of extended postdoctoral training must also be put in balance with a form of survivor's bias, where those that can afford the cost of long-term postdocs tend to exhibit characteristics correlated with the odds of securing a permanent position[38]. Long-term postdoctoral trainees typically fit the profile of tenured researchers, as they tend to originate less often from under-represented minorities and are more often male[26,39]. The impact of long-term postdoctoral training is then hard to assess in isolation (e.g., women more frequently depart from science after a first postdoc to take care of children because of traditional family structures, cf. Ginther and Kahn[40], Martinez et al.[41]). Our study finds no systematic link between training duration and academic success. However, we can not exclude a potential negative influence of long postdocs that might be masked by confounds like gender or minority status, as these variables were not included in the model.

The academic age of the postdoctoral mentor at the time of training had a small but significant positive impact on the odds of continuing in academia (Fig. 4). In contrast, graduate mentor age was not predictive. Several factors might explain the benefit of a more experienced mentor: greater practical knowledge, more material resources in a stable, well-funded laboratory, and better social connections in the network likely to hire the trainee. Future studies controlling specifically for these factors may elucidate their relative importance.

Of interest, the study of Malmgren et al.[19] reported opposite effects for the field of mathematics. For mathematicians, the age of the graduate mentor was negatively correlated with trainee proliferation. This difference from the positive effect that we observe for mentor age in life sciences may arise from the different populations of researchers. Our study considered scientists trained in recent decades, whereas Malmgren et al.[19] limit their analyses to PhD students graduating between 1900 and 1960. Our study also focused on researchers trained by both a graduate and postdoctoral mentor rather than dyads formed by just a graduate advisor and student. Finally, the academic opportunities of mathematics graduates are generally thought to be more advantageous than in biosciences[27,42], making direct comparison between the two datasets difficult. Regardless of the cause, these differences suggest that the features of good mentors vary depending on the broader social and academic context. It may be that effective mentoring strategies and profiles depend on the job market faced by trainees.

Quantitative analysis of mentorship networks has the potential to reveal how multidisciplinary research evolves from mentors in disparate fields[43] and to characterize features of mentorship that lead to successful trainees[19]. Large datasets supporting this type of analysis have not been available historically, but two major databases have been developed recently for this purpose: The Mathematics Genealogy Project (MGP, genealogy.math.ndsu. nodak.edu) and the Academic Family Tree (AFT, used in this study, academictree.org). While these projects both collect mentorship data, they differ in their implementation. The MGP contains roughly 230,000 entries, specifically in the field of mathematics, while the AFT contains about 700,000 entries, distributed across 61 fields, ranging across neuroscience (the first field documented and still the most numerous), chemistry, philosophy, and history. The MGP documents exclusively PhD advisorships, which are the predominant relationship in mathematics. The AFT documents several types of relationship: graduate student, postdoctoral trainee, and staff scientist, reflecting the diversity of training relationships that is more common in other fields, particularly biosciences. The MGP is manually curated (all edits are reviewed by its core team), whereas the Academic Family Tree is largely crowdsourced, with curation based on voluntary user reports. Thus, while the scope of the AFT makes it more relevant to the questions posed in the current study, its broad focus and foundation on crowdsourcing make its sampling less complete than the MGP. There are also biases in its coverage. Some fields, such as neuroscience, are much better represented than others. Here, we showed that analyses for other life sciences disciplines are not in conflict with the results for neuroscience, and that the core intellectual synthesis effect has general validity in life science. However, the claim that all the other patterns studied here hold across all life science fields appears premature at this point, pending a larger dataset. Also, more recent, active researchers may be represented more completely than researchers from earlier decades. Additionally, the AFT over-represents researchers from the United States, although other countries are picking up the pace of filling in data. The current study included several controls to account for possible sampling bias, but as the project continues to grow, the more complete dataset should improve the accuracy of analysis.

Whether they focus on bibliometry or academic genealogy, studies in the science of science aim to uncover individual factors of academic success and attempt to treat broad temporal trends as nuisance factors. However, as this study demonstrates, excluding temporal dynamics is difficult. Funding patterns change, fields grow and contract, and the training patterns of new researchers evolve. Long-term changes in reading and citation patterns have been established[44], indicating that basic features of the scientific production process are changing. The rise of automated data-mining technologies predicts even larger changes in the future. In this context, documenting historical trends, such as funding patterns from major government bodies and the growth of different academic subfields, should provide useful extensions of academic genealogy datasets. In addition to temporal data, incorporating additional data about researchers, such as publications, funded research projects, and collaborations, will provide a more comprehensive understanding of how mentorship impacts scientific research.

## Methods

**Data preparation**. The goal of the modeling effort was to assess and predict success in academic research careers. Many different notions of success can be put under this umbrella. In this study, we chose to quantify trainee proliferation, the number of scientists trained by a scientist, as the measure of academic success[18,19]. Data for the current study were drawn from the Academic Family Tree, an online, crowdsourced database of mentoring relationships. The database records the identity of the mentor and trainee, the type of relationship (graduate or

postdoctoral), and the start and end year of the relationship. Researchers are also linked to publications they have authored that are listed in the Medline database, using a semi-automated algorithm based on string matches to their name and the names of associated trainees and mentors[45]. Because each trainee can themselves be a mentor for subsequent trainees, the database is represented as a directed graph tracing the growth of academic fields across multiple generations of researchers (Fig. 1a). In order to normalize proliferation measures across mentors and trainees who might still be at different stages of their careers, we computed *proliferation rate*, the average number of trainees per decade since becoming an independent researcher.

As of August 2017, the Academic Family Tree dataset contained 670,000+ researchers across more than 30 fields. Data collection for the Academic Family Tree initially focused only on the field of neuroscience[18]. Thus this field is more completely represented than fields added more recently. However, its overall properties are similar to other life science fields, including the mentors proliferation rate (Fig. 7). We thus pooled together all fields of life science in this analysis.

**Selection criteria**. We identified 20,695 triplets in the Academic Tree database, each consisting of a trainee with one graduate and one postdoctoral mentor. In some outlier cases, more than four graduate or postdoctoral advisors were listed, resulting in the same protégé contributing to many triplets. To avoid overcounting these trainees, we removed data for trainees with more than four graduate or postdoctoral mentors. This resulted in 18,856 triplets encompassing 12,853 unique trainees, 9111 unique graduate mentors, and 7322 unique postdoc mentors.

**Missing date inference**. Start and end dates of training were entered optionally by users through the web interface. In 49% of the triplets, both start and end dates were available. Training end dates were more relevant to the analyses presented here, as they marked the transition to the status of independent researcher. When both dates were missing, we inferred the end date by identifying the earliest commonly authored publication and adding the field-specific median lag between the start year and first publication. This was performed separately for graduate and postdoctoral training, based on local regression models (LOESS[46]) to account for changes in training duration over time. For trainees without an end date but with a start date, we estimated the missing date information by adding the mean difference computed from trainees with complete data, again adjusting for changes in the duration of training periods over time. Using this approach, we were able to assign end dates for training period for 90% of the graduate and 89% of the postdoctoral relationships (see Supplementary Table 1). Overall, 15,583 triplets (83% of the triplets) had dates that could be fully inferred, both at the graduate and post-doctoral levels.

**Semantic analysis**. Academic Family Tree researchers were linked to publications in the Medline and Scopus databases using a simple disambiguation procedure. Candidate publications for a researcher were identified by a simple string match between their name and an author name. Each candidate publication was then classified as a high- or low-probability match based on several factors: clusters in the co-author network[45], co-author name matches to adjacent nodes in the mentorship graph (i.e., the researcher's mentor or trainee), and Scopus author identifiers matching other high-probability publications for that researcher. Website users could then curate publication attributions. As of August 2017, 19.3 million publications have been scanned by the automated system. 197,736 publications for 6607 researchers have been curated by users, in 90% agreement with the automated system. 23,709 of the 621,577 publications from the subset of researchers included in the final model were manually curated. On this subset of publications, the agreement between automatic and curated matches reached 93%.

For publications linked to authors, we performed a latent semantic analysis on abstract text to produce a 400-dimensional vector representing semantic content of each researcher's publications (cf. Fig. 2b and Achakulvisut et al.[22], Deerwester et al.[47], Hofmann[48], Pedregosa et al.[49]). Prior to dimensionality reduction, abstracts were pre-processed with stemming, rare word removal, and English stop words removal using the Science Concierge tool suite[22]. The vector space was then generated by applying a term-frequency inverse document frequency (TF-IDF) transformation to abstracts for about 90,000 authors, followed by truncated singular value decomposition to the 400 dimensions with greatest variance across authors. Semantic similarity between two researchers was computed by the cosine distance between the vector average across their publications (Fig. 2c). This metric of semantic similarity has been validated previously[22]. We also compared the semantic similarity metric to a coarser measure based on overlap of keywords for publications in the PubMed database. When we compared the fraction of overlap of PubMed keywords for 450,000 pairs of authors to the semantic similarity of their publications, the correlation coefficient between these metrics was 0.70.

For computing similarity between two mentors in a triplet, we included only publications prior to any co-authored publication with the trainee. For the predictive model of trainee success, we computed graduate mentor/trainee and postdoc mentor/trainee similarities based only on non-coauthored publications

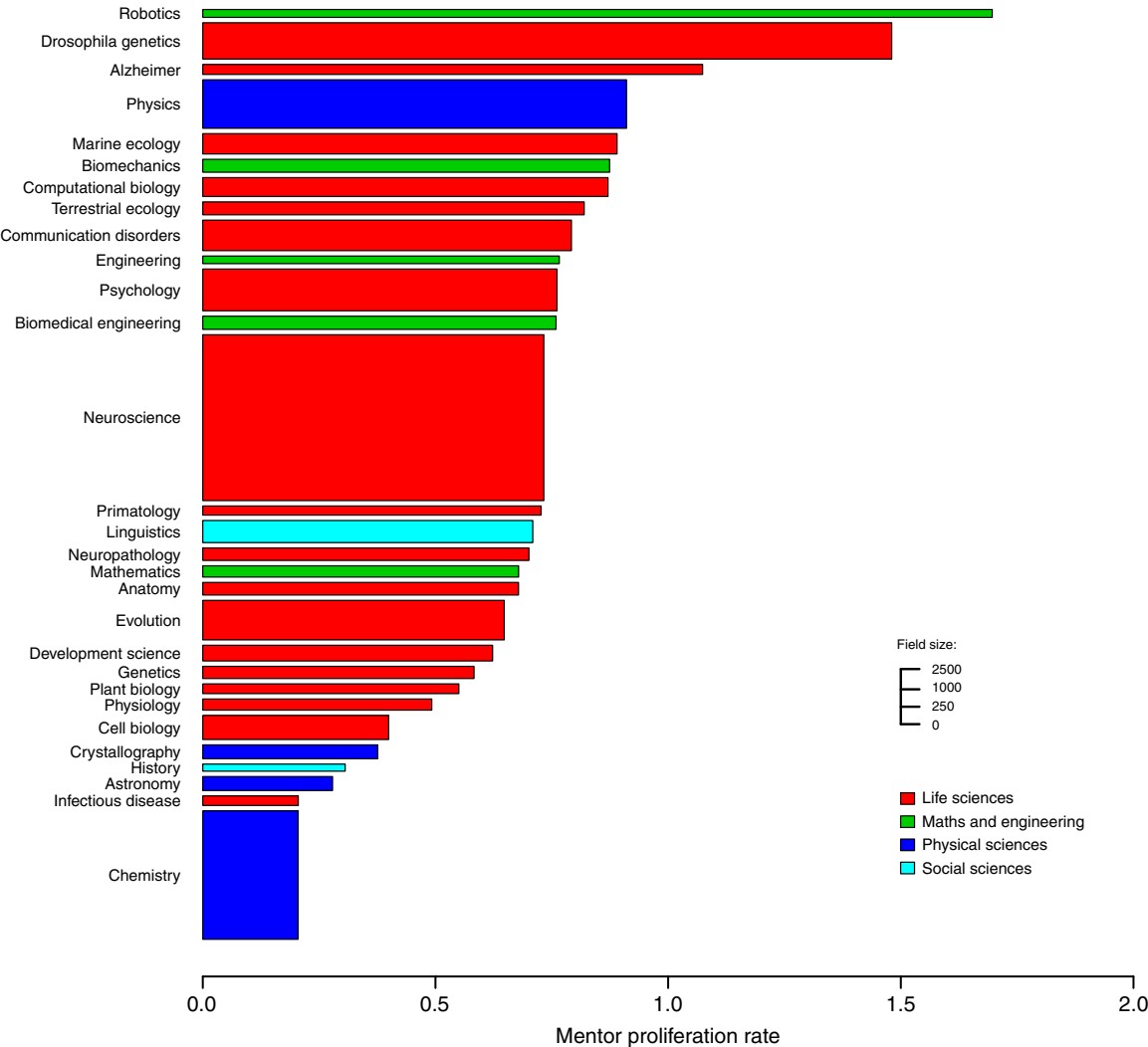

**Fig. 7** Statistical mode of the mentor proliferation rates (horizontal axis) and sample size (number of triplets along the vertical axis). Life Sciences (red) are well-represented in the Academic Family Tree dataset, and thus were the focus of the current study. In most sub-fields, the annual proliferation rate is between 0.5 and 1 (one new trainee every year or every other year). Maths and Engineering fields (green) show generally higher proliferation rates, while Physical Sciences (blue) and Social Sciences (cyan) show lower rates

before the end of postdoctoral training. Thus all publication variables were computed from data available before the end of training.

**Predictors of success**. We hypothesized that several network and semantic variables could predict trainees success:

- training end year: year a trainee completed their last postdoctoral fellowship.
- postdoc duration: total number of years of postdoctoral training.
- graduate mentor proliferation, postdoc mentor proliferation: the average number of trainees per decade for the graduate or postdoctoral mentor, computed the same way as the trainee proliferation rate.
- mentor graph distance: minimum number of steps to pass through a common ancestor between graduate and postdoctoral mentors in the mentorship graph (Fig. 1a).
- graduate mentor age, postdoc mentor age: number of years since the mentor completed their own training ("academic age" in ref. [32]).
- mentor publication similarity: publication similarity (cosine distance between average publication vectors) between mentors for papers published before they started training the protégé and excluding any co-authored publications.
- graduate mentor/trainee similarity, postdoc mentor/trainee similarity: publication similarity between trainee and mentor for publications before the end of postdoctoral training and excluding any co-authored publications.
- publications with graduate mentor, publications with postdoc mentor: number of co-authored publications between mentor and trainee prior to the end of postdoctoral training.

To avoid bias in the similarity measure due to co-authored publications (which would have artificially increased publication similarity), we specifically excluded them in the publication similarity computations. That is, graduate mentor/trainee similarity was computed using publications where they do not appear as co-authors. In practice, the publication corpus of the trainee was thus mostly composed of publications co-authored with the postdoctoral advisor.

Data for some variables was only sparsely available. In particular, mentor/mentor and mentor/trainee publication similarity could be computed only for 50% of triplets, as it required identifying publications by each researcher. Likewise, mentor academic age was not widely available, and could be inferred only for 51% of the triplets (see criteria for inference above). Typically, these variables were harder to identify for mentors, as publication data are limited for earlier dates in the Medline and Scopus databases. Thus, of the total number of 18,856 triplets identified in the database, 15,363 had complete temporal information (training end year and postdoc duration), 6210 had a complete set of mentorship network variables (mentor proliferation, age, and graph distance), and 9513 had a complete set of semantic variables (mentor/mentor and mentor/trainee similarity and number of co-authored publications). Overall 4157 triplets satisfied the criterion for a complete dataset. From this set of triplets, trainees who completed their training after 2007 were excluded from modeling to take into account the time needed to train students or postdocs when continuing in academia, resulting in 1345 triplets that could be used to screen the impacts of all factors. The exclusion of non-significant factors (Table 1 in Results) increased the number of triplets available to fit the final model to 2109.

**Model framework**. Only a fraction of academically trained individuals go on to have an academic career, and those who do not pursue an independent academic career generally do not have an opportunity to train someone. Thus, overall trainee proliferation depends on two factors: first, whether the postdoctoral researcher secured a permanent position with the opportunity to train new researchers and second, how many individuals they trained during their subsequent academic career. To account for these two possibilities, we adopted a zero-inflated model formalism. In this framework, the probability of continuing in research is modeled by a binomial variable, and the proliferation of researchers that moved on to a permanent research position is modeled by a count variable. Given a vector of predictor variables, $X$ (Section), the model simultaneously describes $\pi(x)$, the probability of continuing in an academic career after postdoctoral training, and $f(X)$, the expected number of trainees for those who do continue, as:

$$\log(\pi(X)) = \alpha + \sum \beta_i X_i \quad (1)$$

$$\log(f(X)) = \gamma + \sum \delta_i X_i + \log(C) \quad (2)$$

Parameters $\beta_i$ and $\delta_i$ indicate the relative weight of the $i^{th}$ variable in predicting $\pi$ and $f$, respectively. The career length is introduced as an offset, $\log(C)$, because we are ultimately interested in comparing training rates of mentors. The career length is computed as the difference between the year training was completed (end of the last postdoctoral fellowship) and the current year, capped at 45 years (the longest career length reported in the database). Note that in contrast with the usual statistical convention, we define $\pi(X)$ as the probability of continuation and not the probability of zero-inflation, which is $1 - \pi(X)$[50].

The interpretation of coefficients differs from standard linear regression due to the presence of the log-link ([51], ch. 3–4). A change of one unit in the predictor $X_i$ corresponds to multiplying the chance of continuing an academic career by $\exp(\beta_i)$ (log-binomial model) and multiplying the expected trainee proliferation by $\exp(\delta_i)$ (count model). The exponentiated values of the intercepts, $\exp(\alpha)$ and $\exp(\gamma)$, respectively, indicate the baseline continuation probability and proliferation.

In this modeling framework, it is assumed that any researcher who has trained at least one individual has continued in an academic career. However, researchers without a trainee have not necessarily ended their academic career. Such a modeling choice is well-suited to count data with zero-inflation. It is preferred over a simpler linear regression for the following reasons: (a) two distinct processes leading to the absence of trainees are modeled explicitly, (b) correct boundary conditions are enforced by the model design (i.e., the risk of stopping one's academic career is guaranteed to fall in the range [0,1] and trainee proliferation is never negative) and (c) the number of trainees is not assumed to be normally distributed and can display the over-dispersion expected with count processes.

To confirm that our choice of model formulation and predictors was appropriate, we compared its goodness-of-fit against several alternative formulations (hurdle and zero-inflated, with Poisson and Negative Binomial count models) and predictor sets (Supplementary Table 2). For each configuration, we evaluated the predictive log-likelihood, computed on held-out data that was not used for fitting[52–54]. This cross-validation framework is useful for comparing models that do not assume normally distributed errors and that differ in their number of free parameters[54]. More specifically, we used $k$-fold cross-validation[53], where the data is split into $k$ equal-sized random folds, $y_1,...,y_k$. We define $\theta^{-j}$ as the model parameterization ($\beta$ and $\delta$ from Eq. 1) fit by maximizing log-likelihood on all folds except $y_j$. The predictive log-likelihood in cross-validation $\mathcal{L}$ is then,

$$\mathcal{L} = \sum_{j=1}^{k} \log(p(y_j|\theta^{-j})) \quad (3)$$

The final estimate of $\mathcal{L}$ is derived as the average of 100 iterations made each on a different random 10-fold partition. The zero-inflated and hurdle models were optimized using the maximum likelihood procedure implemented in the R package "countreg"[50].

**Ranking predictors**. Shapley values provide an unbiased assessment of the contribution of individual predictor variables to model performance when they are not entirely independent. This metric was originally developed in the field of game theory to score the contribution of each player (here, predictor variable) to coalitions[25]. The Shapley value is computed by considering all possible combinations of predictors and observing how changing the predictor composition alters model performance (here, cross-validated log-likelihood). Formally, given the log-likelihood $\mathcal{L}$ and the set of predictors $M$, the Shapley value $\zeta_i$ of the predictor $i$ is defined as:

$$\zeta_i(v) = \sum_{S \subseteq M \setminus \{i\}} \frac{|S|!(|M|-|S|-1)!}{|M|!} (\mathcal{L}(S \cup \{i\}) - \mathcal{L}(S)) \quad (4)$$

In regression, a similar approach has been developed to quantify the relative importance of regressors by averaging the goodness-of-fit over all possible combinations of variables[55,56], resulting in a rediscovery of the Shapley values[57]. Extending this approach, Cohen et al.[58] proposed an alternative way to use Shapley values in the context of classification, by using it in an iterative variable selection algorithm. Their Contribution–Selection Algorithm (CSA) has a forward and a backward version, which consist in iteratively adding (or removing for the backward version) the variable with the best (or worst) Shapley value.

In this study, we computed the cross-validated log-likelihood on the entire set of permutations using all or a subset of the variables. We then considered four ways to select the relevant variables to quantify the odds of continuing in academia and the proliferation rate when continuing in academia, namely: with a brute-force approach (picking the combination maximizing cross-validated log-likelihood); with Shapley value computed on the full set of possible combinations (as in ref. [56]); and with the forward and backward versions of the CSA ([58] see Table 1).

**Sensitivity and uncertainty analysis**. We computed 95% bootstrapped confidence intervals for descriptive statistics of the dataset and model prediction[59]. They are shown as error bars and shaded areas throughout the figures.

## Data availability

Data from the Academic Family Tree is licensed for re-use with attribution (CC-BY 3.0) and is available through the web portal https://academictree.org or upon request to davids@ohsu.edu.

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

## Acknowledgements

This project was funded by a NSF EAGER Award (#1646635) and a Metaknowledge Network Grant to Stephen V. David, and AWS Cloud Credits for Research to Jean F. Liénard. Daniel E. Acuna was funded by the National Science Foundation awards #1646763 and #1800956.

## Author contributions

J.F.L. designed and performed all analyses, prepared the initial manuscript and addressed the reviews. T.A. and D.E.A. designed the latent semantic analysis of abstracts. S.V.D envisioned the project, acquired funding, and designed all analyses. All authors contributed to editing the paper.

## Additional information

**Competing interests:** The authors declare no competing interests.

