## [Peer Review File · Nature Communications]

Reviewers' Comments:

Reviewer #1:

Remarks to the Author:

Lienard et al make the bold claim that successful intellectual synthesis by trainees increases their chances of obtaining independent positions. I firmly believe that this manuscript addresses a question of major importance for the scientific enterprise and that the manuscript has the potential to be of interest to the broad readership of Nature Communications.

Sadly, I do not believe that the manuscript is yet technically sound enough. Let me explain, the main finding of the manuscript makes perfect sense and agrees with my intuition. I suspect it confirms the authors' intuitions too. This is a sign that we all should be extra careful not to be too easily convinced.

Reading through the manuscript, there are a number of things that make me uncomfortable.

First, the analysis extends over a long period of time over which many things changed — NIH funding doubled and then stopped growing, neuroscience increased dramatically, computational biology was born, etc. In spite of this, the authors do not attempt to segregate their data in ways that would remove potential confounding factors. And, no, the results of Fig 8 are not convincing!

Second, data availability is likely to be grossly inconsistent in terms of coverage of different fields. The data in Figure 7 suggest that others are likely poorly covered when compared to neuroscience. By the way, the values on the x-axis for "mentor proliferation rate" do not match the values mentioned in the caption.

Third, there is no indication that triplets were restricted geographically at all. However, it is likely that career opportunities and coverage vary widely with geography and pedigree (institutional affiliation).

Fourth, the definition of similarity for trainees and mentors is presented but not supported. Why should I trust the metric presented in Figs 1B,C? Why not assign topics to the publications and look at a vector over topics instead of over terms? The projection on terms is likely very very noisy.

I believe that all these issues are easily addressable by the authors. They do not need to get any new data and likely only need to write some more code.

Reviewer #2:

Remarks to the Author:

In this paper, the authors conduct an analysis of a database of about 20,000 researchers in biomedical sciences, focusing on individuals with both a graduate degree and postdoctoral training recorded in the database. (They use the term "triplets" to describe these structures, but the definition, as best as I can tell, appears only in the caption of Figure 2, which is not until after the term has been used multiple times.) They look at a variety of features of the mentorship network in order to determine what increases the success of a protege. (The authors define success as training either a graduate student or postdoctoral fellow with a record in the database under study.) The authors' principal conclusion is that what they term "intellectual synthesis" is the strongest of the factors they studied at influencing continued academic research. Their conclusion is based on an analysis of abstracts of papers written by the individuals in the database, using a stemmed keyword approach. They find that proteges who publish papers involving keywords in common with their doctoral advisor and their postdoctoral mentor are more successful, provided those two keyword sets have some but not too many commonalities. The authors conclude that in

these situations, the protege has synthesized the work going on in two loosely-related disciplines and increased their chances of success. The authors also find some connection between an individual's success and the rate at which their mentors train proteges and the experience level of the postdoctoral mentor (but not doctoral mentor).

The authors' work is novel and fits into the existing literature of similar analyses conducted on other data sets. It should promote further discussion and study of the role of academic networks on researchers' career paths, particularly because the authors have reported their results on a wide variety of possible factors that were examined and found to have little or no impact on an individual's success. I am not qualified to comment on the statistical methods used by the authors, but I applaud them for examining a number of factors and reporting the conclusions they were able to draw from them (or that no reliable connection was found). (I will note that the content of Figure 4 disagrees with the caption, with one mentioning an 8.9% gain and the other 9.3% for the same variable.)

The primary recent work that the authors cite and compare to, particularly with regard to mentor experience, is the work of Malmgren et al. (ref. 34), which looked at the impact of doctoral mentor experience in mathematics by looking at data from the Mathematics Genealogy Project. Malmgren et al. analyzed data from the first 60 years of the 20th century, and because of the content of the Mathematics Genealogy Project database, were restricted to studying only the doctoral advising relationship. In contrast, the authors here have a dataset consisting primarily of individuals who completed their academic training since 1990 and only considered individuals who had postdoctoral training when conducting their analysis. With regard to the mentor experience analysis discussed on lines 307--321, it would be interesting to see if a change to consider dyads as in the paper of Malmgren et al. impacts the conclusions in this paper, since the restriction to individuals with postdoctoral training that took place before the analysis may have an influence. Also of concern in the authors' approach here is that, even after conducting a significant amount of data inference to correct for missing data, mentor academic age was not available for 49% of triplets. The data used by Malmgren et al. would have had much more complete information. The Mathematics Genealogy Project dataset is certainly different to the one considered by the authors here, since it does not include information on postdoctoral mentorship. However, it is a more robust and curated database than that studied by the authors. (For example, restricting the math dataset to just doctoral degrees awarded in the 1990s would result in over 42,000 mentor-protege dyads to study.)

I would also like to see the authors expand the information on the semantic analysis portion of their paper, since that is where their principal conclusion lies. In particular, the author identification used to link researchers to publications were primarily done automatically. The authors cite a 90% agreement rate for the user-curated data with the automated matching, but there is no discussion of what that rate is for the researchers considered in the data set studied in this paper. This is another area where the Mathematics Genealogy Project dataset is more robust than the one under study, since research identification with MathSciNet is quite curated. Although the MGP-MathSciNet identification also involves automation, the MathSciNet database of publications is entirely curated with significant attention paid to author identification by skilled bibliographers. Unfortunately, the lack of postdoctoral mentor data in the Mathematics Genealogy Project dataset means that the authors intellectual synthesis analysis cannot be replicated.

This paper should fuel further discussion and study into what conclusions can be drawn about the role of mentors on researchers' academic careers, and perhaps even encourage the creation of more robust datasets (crowdsourced or curated) to enable further studies. However, there are some shortcomings in terms of the restrictions placed upon the dataset studied relative to prior studies and other datasets that could be investigated in a similar manner to at least some of the authors' work here. (In some cases, such as the author identification, this may not be a shortcoming of the dataset but just a lack of detail in what the authors have provided about their subset of the overall dataset.)

Mitchel T. Keller
Managing Director, Mathematics Genealogy Project

Reviewer #3:

Remarks to the Author:

This paper tackles the question on how mentorship determines chances of a success in scientific careers. The question is addressed by performing a statistical analysis and modelling of a dataset of academic genealogy in the life sciences, where for a number of scientists graduate and postdoctoral mentorship relations are known. The authors use the number of scientific trainees a scientist has during his/her career as a proxy to gauge success. Then the analysis shows the role of different variables, some related to career characteristics of trainees and mentors, like duration of postdoctoral training or academic age of mentors, other related to network/relational properties, like finding common ancestors between graduate and postdoctoral mentors, in determining success.

There are a number of interesting findings in the paper, the most novel ones - as pointed out by the authors - being the fact that the postdoctoral mentor proliferation (number of trainees of the postdoctoral mentor) are a greater effect than the graduate mentor on the odds of getting a permanent position, and that successful scientists are those that have done research similar to their mentors, while the mentors have dissimilar research.

The core of this paper is a thorough and detailed analysis of the dataset mentioned above. Everything looks technically correct and the language is clear, but the paper reads at times like an applied statistics or econometric paper, to the point of sounding aseptic. I feel this is more than just a matter of style, as it limits the reader - especially if from the research policy or science of science community - to go ahead and look into the findings and their implications. This aspect of the paper is also reflected on the references: the authors cite a lot of statistics papers, which I appreciate because we do need sound and robust statistical analysis, but the paper lacks many references of science of science and research policy, and relative discussion. For example, since the paper deals with modelling academic careers, reference and connections to the findings of papers about data-driven models of success in scientific careers would have been expected (see for example, Petersen et al. "Persistence and uncertainty in the academic career", PNAS (2012), Sinatra et al. "Quantifying the evolution of individual scientific impact", Science (2016), Way et al. The misleading narrative of the canonical faculty productivity trajectory, PNAS (2017)). Also I would have expected at least some comments if not an analysis of variables for which there is consensus that they are fundamental for scientific success, like collaboration patterns or teams (for a review of the field, with many relevant references, Fortunato et al. "Science of Science", Science (2018)).

I have then two major issues regarding the analysis and interpretation of some of the results:

- one issue regards the limited temporal dimension of the analysis, with important implications on the interpretations of the presented results. It is not clear whether there is an overall trend for the scientists that are more successful to go from a graduate mentor to a "better" postdoctoral mentor. If such a trend exists, then the statement/explanation that "postdoctoral mentor has a greater effect than the graduate mentor on the odds of securing a permanent position" should be definitely revisited: it might simply be that those scientists that (randomly?) move to a better postdoctoral mentor (measured with proliferation rates) have then better access to opportunities and are more likely to be successful themselves.

- the other issue is about the important finding, also summarised in the title, that successful scientists are those that make a synthesis between two different line of research, represented by

graduate and postdoctoral mentors, with important implications about bridging different areas of knowledge. My issue with this is that again we have no insight about the temporal evolution of the career of the scientist: can it simply be that scientists who move to a more thriving line of research or subfield for the postdoctoral training have more chances to get a permanent position? From this point of view the higher success rate is not due to the intellectual synthesis of the trainee, but to the fact that s/he changed topic and moved to a research area with more opportunities.

The two hypotheses above can be ruled out with the data at hand, and in general more explanations and intuitive insights should be given on why the authors observe what they observe. In general I do believe that in a study of careers one needs to look into and understand the temporal evolution of variables and understand how these connect to the probability of staying in academia and of achieving success. Some of the consequences/explanations I am suggesting might be possible already with the analysis at hand, but if it is the case, they are too much hidden behind the technical details.

In summary, the authors have clearly put a lot of effort into the analysis and have done a great job with the technical details, but the main results, their consequences and the overall narrative is too much hidden by the technical nature of the presentation. Also I think it is fundamental to make a better job in offering more support and clearer explanations for some of the main observations of the paper. I think these issues can definitely be fixed with a thorough revision of the paper.

Minor points:

- References 33 and 34 are duplicates, and the correct one to be used - to my knowledge - is 33 (not Nunes but Amaral)
- line 58: intellectually - intellectually
- line 66 - 67: there is a parenthesis missing, and the sentence is somehow messed up.
- line 94: "about about"
- page 6, figure 2: x- and y- label text is too small, same with legend. When the paper is printed is hard to read.
- line 286: We - we
- line 298: "securing a permanent positions" - "securing a permanent position"

Reviewer #1 (Remarks to the Author):

Lienard et al make the bold claim that successful intellectual synthesis by trainees increases their chances of obtaining independent positions. I firmly believe that this manuscript addresses a question of major importance for the scientific enterprise and that the manuscript has the potential to be of interest to the broad readership of Nature Communications.

Sadly, I do not believe that the manuscript is yet technically sound enough. Let me explain, the main finding of the manuscript makes perfect sense and agrees with my intuition. I suspect it confirms the authors' intuitions too. This is a sign that we all should be extra careful not to be too easily convinced.

Reading through the manuscript, that are a number of things that make me uncomfortable.

First, the analysis extends over a long period of time over which many things changed — NIH funding doubled and then stopped growing, neuroscience increased dramatically, computational biology was born, etc. In spite of this, the authors do not attempt to segregate their data in ways that would remove potential confounding factor. And, no, the results of Fig 8 are not convincing!

(R1.1) We agree that long-term trends play a major role in this dataset. We made a substantial effort in writing the original manuscript to convince ourselves that the other factors influencing professional success are not artifacts of these strong temporal effects. This concern seems especially pertinent to the network related variables, such as mentor proliferation rates, which are likely to correlate with overall growth and contraction of fields. It is less clear how these long-term trends might bias effects of intellectual synthesis, but the data are complicated, and the potential for spurious correlations is a concern.

To account for long term temporal effects more completely, and in particular to account for the nonlinearity of time vs. continuation odds in Fig. 9A, we introduced a series of higher-order polynomial terms into the regression model. These additional terms were in fact able to capture some additional variance in the data, but they had no large impact on the other factors of interest in the regression. In particular, the intellectual synthesis effects appeared robust to the inclusions of finer temporal controls. See figure below.

We also tried another type of non-linear modeling of temporal effects, by replacing our continuous time variable (“Training end year”) by an ordinal variable capturing the same temporal subdivisions as in Fig 8 (“Training end epoch”, numbered from 1 to 5). This transformation did not affect which coefficients were significant nor did it change the order of magnitude of their effects. See figure below.

We have included a discussion of these alternative and more complex temporal models in the text, although we have chosen to keep the simpler model in the main text because it captures the same network and synthesis effects (mark 10 on page 12 and mark 26 on page 8). In addition, we have included a more general discussion of the complexity of this data set and the potential how larger datasets might address concerns about long-term trends and sampling bias (mark 21 on page 19).

As an aside, we also computed the potential influence of NIH funding after reading your remark. Although econometry is not our area of research, we followed the procedure used in⁶ and were able to recompute the graphs therein. After feeding the total NIH funding of the year of last postdoc (in constant

2009 dollar adjusted for inflation in biomedical research, using the BRDPI price index) as a variable to the model, we found that it mediated no significant impact in the odds of continuing in research. See graph below. The only effect of this variable is an interaction with the general temporal variable “Training end date” on long-term proliferation, possibly as a collinear balancing artifact (one variable becomes a significant positive factor while the other becomes a significant negative factor). Our understanding from this analysis is that a complete understanding of temporal effects will require an in-depth analysis incorporating multiple factors, and that generally, the effects of intellectual synthesis appear robust to inclusion of extraneous variables. We do not mention this analysis in the manuscript due to its inconclusive nature.

2009 constant dollar adjusted for BRDPI

Sources used: Biomedical Research and Development Price Index obtained from <https://officeofbudget.od.nih.gov/gbipriceindexes.html>, NIH total funding from <https://www.nih.gov/about-nih/what-we-do/nih-almanac/appropriations-section-2>

Second, data availability is likely to be grossly inconsistent in terms of coverage of different fields. The data in Figure 7 suggest that other are likely poorly covered when compared to neuroscience .

(R1.2) We agree that the strong representation of neuroscience could bias the broader bioscience results. To test for this possibility, we performed a new analysis on the data. We broke the data into two groups: neuroscience only and all fields except neuroscience (split based on whether the trainee was listed in neuroscience) and fit the same regression model on these separate datasets. This analysis revealed similar results in both datasets. There are some differences between them, but the main intellectual synthesis effects are preserved. The results are included in a set of new supplemental figures in Supplementary Information (mark 27 on page 13) that are presented in main text (mark 13 on page 14).

By the way, the values on the x-axis for “mentor proliferation rate” do not match the values mentioned in the caption.

(R1.3) Fixed, thanks.

Third, there is no indication that triplets were restricted geographically at all. However, it is likely that career opportunities and coverage vary widely with geography and pedigree (institutional affiliation).

(R1.4) While we did not include explicitly institutional affiliations as a factor in the current study, we include many other variables correlating with their beneficial/antagonist effects: mentor’s seniority, mentor’s trainee proliferation (this metric is correlated with lab size and has been shown in previous works to be correlated with awards, such as membership in the US National Academy of Sciences), number of co-publications (linked to the overall productivity of mentors). Thus, we already control for many of the factors underlying the beneficial/antagonist aspects associated with prestigious/unknown institutional affiliations.

As for geography, we agree that some sort geographical factors could correlate with the odds of obtaining a permanent position. Indeed, as a sizable portion of life science postdocs in the US have been trained in another country at some earlier point of their life, and we could expect that international mobility has some impact on academic success, in terms of obtaining a position in the US, for example, or in another country. However, this process is fraught with non-trivial technical difficulties, as some of the key variables of interest (nationality and visa situation) are simply out of reach. Also, it is a very intricate and potentially divisive issue, and we believe that treating it as an add-on to this study can not possibly be done in a satisfying way.

To address this concern, we chose to be more thorough in our interpretation of our results. We specifically tried to be more explicit about our thought process and the underlying assumptions in the introduction (mark 2 on page 3) and discussion (mark 20 on page 19).

Fourth, the definition of similarity for trainees and mentors is presented but not supported. Why should I trust the metric presented in Figs 1B,C? Why not assign topics to the publications and look at a vector over topics instead of over terms? the projection on terms is likely very very noisy.

(R1.5) The dimensionality reduction applied to the semantic space can actually be thought of as a way of measuring topics. We initially analyzed the overlap of PubMed keywords. Upon working with our co-authors who specialize in these approaches, however, we found that the current semantic analysis provided a highly correlated but much finer-grained measure of similarity. This particular method for latent semantic analysis has been validated and is the subject of a publication (see Fig. 4 of Achakulvisut et al., 2016). We have expanded our discussion of the relationship between LSA and more traditional measures of semantic similarity in the Methods (mark 24 on page 23).

I believe that all these issues are easily addressable by the authors. They do not need to get any new data and likely only need to write some more code.

Reviewer #2 (Remarks to the Author):

In this paper, the authors conduct an analysis of a database of about 20,000 researchers in biomedical sciences, focusing on individuals with both a graduate degree and postdoctoral training recorded in the database. (They use the term "triplets" to describe these structures, but the definition, as best as I can tell, appears only in the caption of Figure 2, which is not until after the term has been used multiple times.)

(R2.1) Thanks, fixed (mark 4 on page 4).

They look at a variety of features of the mentorship network in order to determine what increases the success of a protege. (The authors define success as training either a graduate student or postdoctoral fellow with a record in the database under study.) The authors' principal conclusion is that what they term "intellectual synthesis" is the strongest of the factors they studied at influencing continued academic research. Their conclusion is based on an analysis of abstracts of papers written by the individuals in the database, using a stemmed keyword approach. They find that proteges who publish papers involving keywords in common with their doctoral advisor and their postdoctoral mentor are more successful, provided those two keyword sets have some but not too many commonalities. The authors conclude that in these situations, the protege has synthesized the work going on in two loosely-related disciplines and increased their chances of success. The authors also find some connection between an individual's success and the rate at which their mentors train proteges and the experience level of the postdoctoral mentor (but not doctoral mentor).

The authors' work is novel and fits into the existing literature of similar analyses conducted on other data sets. It should promote further discussion and study of the role of academic networks on researchers' career paths, particularly because the authors have reported their results on a wide variety of possible factors that were examined and found to have little or no impact on an individual's success. I am not qualified to comment on the statistical methods used by the authors, but I applaud them for examining a number of factors and reporting the conclusions they were able to draw from them (or that no reliable connection was found). (I will note that the content of Figure 4 disagrees with the caption, with one mentioning an 8.9% gain and the other 9.3% for the same variable.)

(R2.2) Thank you, we have corrected this discrepancy to reflect the actual value of 8.9% in the caption (mark 9 on page 12).

The primary recent work that the authors cite and compare to, particularly with regard to mentor experience, is the work of Malmgren et al., which looked at the impact of doctoral mentor experience in mathematics by looking at data from the Mathematics Genealogy Project. Malmgren et al. analyzed data from the first 60 years of the 20th century, and because of the content of the Mathematics Genealogy Project database, were restricted to studying only the doctoral advising relationship. In contrast, the authors here have a dataset consisting primarily of individuals who completed their academic training since 1990 and only considered individuals who had postdoctoral training when conducting their analysis. With regard to the mentor experience analysis discussed on lines 307–321, it would be interesting to see if a change to consider dyads as in the paper of Malmgren et al. impacts the conclusions in this paper, since the restriction to individuals with postdoctoral training that took place before the analysis may have an influence.

(R2.3) We agree that a comparison to the earlier analysis of Malmgren et al. would be very valuable. However, we should note that while PhDs that did not complete a postdoc are the norm in mathematics, at least for the time periods analyzed in Malmgren et al., performing a postdoc is basically a requirement in the life science careers that we study in our work (a trend that started in the 1970s, thus aligned with the bulk of the Academic Tree subset studied here, cf. reference 43 - the US National Research Council report of 1981). In this context, designing a fair comparison between the role of the graduate advisor of Math PhDs from 1900-1960 and whose early career has been influenced mostly by this advisor, and the graduate advisor of more recent Life Science PhDs which we know are also influenced by their postdoctoral advisor, is difficult.

We still sought to address this question by computing the model on the same trainee subset but using only features from their graduate studies. This tests the hypothesis that the absence of effects of graduate mentor's age is not a confound of postdoctoral training variables, based on the same dataset of PhDs that did a postdoc (because it is representative of the norm in life science).

In this case we still found no significant effect of graduate mentor's age, suggesting a difference between mathematics and bioscience fields. See figure below. We added this additional analysis and supporting figure at mark 18 on page 19 (main text) and mark 25 on page 4 (supplementary materials).

Also of concern in the authors' approach here is that, even after conducting a significant amount of data inference to correct for missing data, mentor academic age was not available for 49% of triplets. The data used by Malmgren et al. would have had much more complete information. The Mathematics Genealogy Project dataset is certainly different to the one considered by the authors here, since it does not include information on postdoctoral mentorship. However, it is a more robust and curated database than that studied by the authors. (For example, restricting the math dataset to just doctoral degrees awarded in the 1990s would result in over

42,000 mentor-protege dyads to study.)

(R2.4) We agree, it is frustrating that the crowd-sourced system used in the Academic Family Tree does not require entry of training dates. We chose to be conservative in our method for measuring training dates because we observed that long-term trends have had such a strong influence on trainee success rates. Generally we were able to infer training dates based on publication information. Since publication data were also required for the analysis of intellectual synthesis, we decided to focus on this smaller subset of researchers.

More generally, we agree that a comparison with Malmgren et al. is highly relevant, and we have expanded this part of the discussion to consider differences in data sampling, in addition to the network analysis results already discussed (mark 19 on page 19).

I would also like to see the authors expand the information on the semantic analysis portion of their paper, since that is where their principal conclusion lies. In particular, the author identification used to link researchers to publications were primarily done automatically. The authors cite a 90% agreement rate for the user-curated data with the automated matching, but there is no discussion of what that rate is for the researchers considered in the data set studied in this paper. This is another area where the Mathematics Genealogy Project dataset is more robust than the one under study, since research identification with MathSciNet is quite curated. Although the MGP-MathSciNet identification also involves automation, the MathSciNet database of publications is entirely curated with significant attention paid to author identification by skilled bibliographers. Unfortunately, the lack of postdoctoral mentor data in the Mathematics Genealogy Project dataset means that the authors intellectual synthesis analysis cannot be replicated.

(R2.5) This is a helpful point. We have revised the text to cite the study that developed and validated the semantic analysis method (mark 24 on page 23). We also now report accuracy of author identification for the subset of researchers included in the paper, which is 93% (mark 23 on page 23).

This paper should fuel further discussion and study into what conclusions can be drawn about the role of mentors on researchers' academic careers, and perhaps even encourage the creation of more robust datasets (crowdsourced or curated) to enable further studies. However, there are some shortcomings in terms of the restrictions placed upon the dataset studied relative to prior studies and other datasets that could be investigated in a similar manner to at least some of the authors' work here. (In some cases, such as the author identification, this may not be a shortcoming of the dataset but just a lack of detail in what the authors have provided about their subset of the overall dataset.)

Mitchel T. Keller Managing Director, Mathematics Genealogy Project

(R2.6) Thanks. We agree that a developing more complete and accurate datasets will be valuable for further study of these issues. We have included points about this future direction, which we plan to pursue, in the revised Discussion (mark 22 on page 20).

Reviewer #3 (Remarks to the Author):

This paper tackles the question on how mentorship determines chances of a success in scientific careers. The question is addressed by performing a statistical analysis and modelling of a dataset of academic genealogy in the

life sciences, where for a number of scientists graduate and postdoctoral mentorship relations are known. The authors use the number of scientific trainees a scientist has during his/her career as a proxy to gauge success. Then the analysis shows the role of different variables, some related to career characteristics of trainees and mentors, like duration of postdoctoral training or academic age of mentors, other related to network/relational properties, like finding common ancestors between graduate and postdoctoral mentors, in determining success. There are a number of interesting findings in the paper, the most novel ones - as pointed out by the authors - being the fact that the postdoctoral mentor proliferation (number of trainees of the postdoctoral mentor) are a greater effect than the graduate mentor on the odds of getting a permanent position, and that successful scientists are those that have done research similar to their mentors, while the mentors have dissimilar research. The core of this paper is a thorough and detailed analysis of the dataset mentioned above. Everything looks technically correct and the language is clear, but the paper reads at times like an applied statistics or econometric paper, to the point of sounding aseptic. I feel this is more than just a matter of style, as it limits the reader - especially if from the research policy or science of science community - to go ahead and look into the findings and their implications. This aspect of the paper is also reflected on the references: the authors cite a lot of statistics papers, which I appreciate because we do need sound and robust statistical analysis, but the paper lacks many references of science of science and research policy, and relative discussion. For example, since the paper deals with modelling academic careers, reference and connections to the findings of papers about data-driven models of success in scientific careers would have been expected (see for example, Petersen et al. "Persistence and uncertainty in the academic career", PNAS (2012), Sinatra et al. "Quantifying the evolution of individual scientific impact", Science (2016), Way et al. The misleading narrative of the canonical faculty productivity trajectory, PNAS (2017)). Also I would have expected at least some comments if not an analysis of variables for which there is consensus that they are fundamental for scientific success, like collaboration patterns or teams (for a review of the field, with many relevant references, Fortunato et al. "Science of Science", Science (2018)).

(R3.1) Thank you for pointing us to this interesting literature. They were helpful to improve the introduction (mark 1 on page 2), and overall to relate our approach focused on academic genealogy to the existing body of bibliometric works.

Also thanks to your feedback on style, we streamlined our presentation of the results to avoid distracting some potential readers. In particular, we opted to some of the most technical aspects of the results about model selection (negative binomial / Poisson distribution and zero-inflated / hurdle model structure) to Supplementary Information, and kept only a brief summarized form in main text (mark 8 on page 9).

I have then two major issues regarding the analysis and interpretation of some of the results: - one issue regards the limited temporal dimension of the analysis, with important implications on the interpretations of the presented results. It is not clear whether there is an overall trend for the scientists that are more successful to go from a graduate mentor to a "better" postdoctoral mentor. If such a trend exists, then the statement/explanation that "postdoctoral mentor has a greater effect than the graduate mentor on the odds of securing a permanent position" should be definitely revisited: it might simply be that those scientists that (randomly?) move to a better postdoctoral mentor (measured with proliferation rates) have then better access to opportunities and are more likely to be successful themselves.

(R3.2) This is a relevant concern that we failed to consider in the original manuscript. To test this possibility, we re-fit the model with a new term "postdoc vs. graduate mentor proliferation", computed as the ratio of postdoc/graduate mentor proliferation rates. This number should identify systematic benefits of moving to a more prolific postdoc mentor. This additional term had no significant predictive power, and thus we infer that the proposed pattern of mentorship is not the common mode for successful trainees. We have included this analysis in a new section of the results (mark 14 on page 15).

- the other issue is about the important finding, also summarised in the title, that successful scientists are those that make a synthesis between two different line of research, represented by graduate and postdoctoral mentors, with important implicants about bridging different areas of knowledge. My issue with this is that again we have no insight about the temporal evolution of the career of the scientist: can it simply be that scientists who move to a more thriving line of research or subfield for the postdoctoral training have more chances to get a permanent position? From this point of view the higher success rate is not due to the intellectual synthesis of the trainee, but to the fact that s/he changed topic and moved to a research area with more opportunities.

(R3.3) We agree that this is a likely model for at least some successful trainees, and agree that it may be difficult to tease apart in the current dataset, as a successful postdoctoral mentor is themself likely to be in a thriving subfield. Thus the concern does not seem to reflect a potential bias as much as a different interpretation of the results.

We did consider one alternative model with a new interaction term, (trainee-postdoc mentor similarity * postdoc mentor proficiency), contrasted with a second term: (trainee-graduate mentor similarity * graduate mentor proficiency). The first term should be high for trainees who moved into the "better" subfield of their postdoc mentor, while the second term contrasts this effect and should be high for trainees who stayed in the "better" subfield of their graduate studies. These terms were not linked to increased (or decreased) odds of finding a permanent position in this dataset. See graph below.

Interestingly, there is a very small effect of the “Postdoc mentor rate x similarity” term on long-term proliferation, which is an effect hard to interpret. It is quite small, and we know that interaction terms of significant variables tend to be significant themselves and may balance the effect of their original variables (in this regression, the effect of the “Postdoc mentor rate” on the number of trainees is indeed larger than in the original regression), so we avoid reading too much into this new effect.

We include a description of these results in the new section on network-semantic interactions (mark 15 on page 15). We have also included a discussion of different models for postdoctoral mentor influence in the discussion (mark 16 on page 17), commenting that the specific benefits of a successful postdoctoral mentor may be variable but that the main effects of the paper (greater influence of postdoctoral mentor and intellectual synthesis) still hold, despite these alternative mechanisms.

The two hypotheses above can be ruled out with the data at hands, and in general more explanations and intuitive insights should be given on why the authors observe what they observe. In general I do believe that in a study of careers one needs to look into and understand the temporal evolution of variables and understand how these connect to the probability of staying in academia and of achieving success. Some of the consequences/explanations I am suggesting might be possible already with the analysis at hand, but if it is the case, they are too much hidden behind the technical details.

In summary, the authors have clearly put a lot of effort into the analysis and have done a great job with the technical details, but the main results, their consequences and the overall narrative is too much hidden by the technical nature of the presentation. Also I think it is fundamental to make a better job in offering more support and clearer explanations for some of the main observations of the paper. I think these issues can definitely be fixed with a through revision of the paper.

(R3.4) These comments about possible interactions between network and semantic effects are useful for clarifying out thinking and have lead to the expanded Discussion on this topic (mark 16 on page 17).

Minor points:

- References 33 and 34 are duplicates, and the correct one to be used - to my knowledge - is 33 (not Nunes but Amaral)

■ (R3.5) *Fixed.*

- line 58: intellectually —> intellectually

■ (R3.6) *Fixed (mark 3 on page 3).*

- line 66 - 67: there is a parenthesis missing, and the sentence is somehow messed up.

■ (R3.7) *Thanks, fixed (mark 5 on page 4).*

- line 94: "about about"

■ (R3.8) *Fixed (mark 6 on page 4).*

- page 6, figure 2: x- and y- label text is too small, same with legend. When the paper is printed is hard to read.

■ (R3.9) *Thank you, we increased the size of the labels / axes (mark 7 on page 7)*

- line 286: We —> we

■ (R3.10) *Fixed (mark 17 on page 18).*

- line 298: "securing a permanent positions" —>"securing a permanent position"

■ (R3.11) *Fixed, thanks (mark 11 on page 13).*

Intellectual Synthesis in Mentorship Determines Success in Academic Careers.

Jean F. Liénard^{1*}, Titipat Achakulvisut ², Daniel E. Acuna ³ and Stephen V. David¹

¹ Oregon Hearing Research Center, Oregon Health & Science University, Portland, Oregon, United States of America

² Department of Bioengineering, University of Pennsylvania, Philadelphia, Pennsylvania, United States of America

³ School of Information Studies, Syracuse University, Syracuse, New York, United States of America

Corresponding author: * jean.f.lienard@gmail.com

Abstract

As academic careers become more competitive, junior scientists need to understand the value that mentorship brings to their success in academia. Previous research has found that, unsurprisingly, successful mentors tend to train successful students. But what characteristics of this relationship predict success, and how? We analyzed an open-access database of about 20,000 researchers who have undergone both graduate and postdoctoral training, compiled across several fields of biomedical science. Our results show that postdoctoral mentors were more instrumental to trainees' success compared to graduate mentors. A trainee's success in academia was also predicted by the degree of intellectual synthesis with their mentors, resulting from fusing the influence of disparate advisors. This suggests that junior scientists should have increased chances of success by training with and linking the ideas of mentors from different fields. We discuss the implications of these results for choosing mentors and determining the duration of postdoctoral training, 
[revised manuscript text omitted]
 ~~variable~~ mix of research and university-level teaching. ~~This seldom occurs outside of academia, in the private sector.~~ Here, we focus specifically on a criterion that ~~is a good proxy characterizing the~~ provides a proxy for success of academic research careers in life science, and which is accessible in our dataset: the training of at least one graduate student or ~~postdoc~~ postdoctoral fellow. Indeed, the training of a ~~more~~ junior researcher is a years-long commitment, and a stable research position is often an institutional prerequisite for it. Conversely, ~~having postdocs and graduate students is the norm in life science research, and~~ virtually all successful researchers in life science manage a team composed of graduate students and postdocs.

~~To investigate the impact of publication similarity on trainee academic success~~ Using publication similarity as a measure of intellectual overlap between researchers, we considered its relationship to the odds of becoming a mentor, *i.e.*, for the trainee to continue in academia and themselves train at least graduate student or postdoctoral fellow. We observed several significant correlations between publication similarity and trainee success. In particular, greater publication similarity between a trainee and each mentor led to higher probability of continuing in academia (Fig. 3A-B, $p < 0.001$, two-sample Kolmogorov-Smirnov test). In contrast, co-mentors with greater publication similarity had trainees *less* likely to continue in research (Fig. 3C, $p < 0.001$). Together, these observations are consistent with a model in which a trainee who successfully synthesizes knowledge and approaches from dissimilar mentors is more likely to continue on to an independent academic career. Furthermore, successful trainees tended to show closer semantic proximity to the postdoctoral mentor than the graduate mentor (Fig. 3D, $p < 0.001$), suggesting that the postdoctoral relationship is a stronger determinant for the trainee's future employment.

Figure 2: Main features of the $n = 18,856$ mentorship triplets in life science analyzed in this study. Each triplet is constituted by a pair of graduate and postdoc mentors and their common trainee. A: distribution of proliferation rate (average number of trainees per decade) of mentors (blue) and trainees (red). The large peak at zero for trainees reflects the large number of trainees that do not go on to mentor students of their own. B: difference in proliferation rate between graduate and postdoc mentors common to triplets (blue) and mentor pairs picked at random (red). C: common ancestor distance of co-mentors (blue) and mentors picked at random. D: academic age at the time of training for graduate and postdoc mentors of each triplet. E: year of graduation (blue) and end of postdoc (red) for triplets. The decreasing number of trainees graduated in recent years, in blue, results from the inclusion criterion of trainees that were selected as those having undergone postdoctoral training. F: Mean cumulative time spent in graduate studies (blue) and postdoctoral fellowships (red), as a function of postdoc end date. Postdoctoral studies lasting more than 8 years were excluded from the computation of cumulative time. Instead their proportion was plotted as bars at the bottom the graph. G: the similarity between co-mentors (blue) is higher than among a randomly picked pair of researchers (red). H: closer common ancestor distance leads to greater publication similarity, and this effect is cumulative with the higher proximity of researcher that co-mentor the same trainee.

Figure 3: The odds of becoming an academic mentor are correlated with the trainee’s ability to synthesize disparate influences from mentors, as revealed by the similarity of their abstracts published prior to the end of the postdoc. Arrows indicate the medians (the differences are significant in all panels, $p < 0.001$). A-B: Trainees who become mentors showed greater similarity with their graduate (A) and postgraduate (B) advisors. C: Lower similarity between mentor is linked with better odds to continue in academic research. D: Protégés that have a greater publication proximity with their postdoc mentors, compared to their graduate mentors, tend to move more often to independent academic positions.

2.3 Model of academic success in life science.

The patterns in Figs. 2 and 3 suggest a link between the characteristics of mentors and their trainee’s odds of staying in academia. At the same time, the relatively strong coupling between variables such as postdoc duration and training end date (Fig. 2F) and mentor graph distance and publication similarity (Fig. 2H) presents a challenge for building the predictive model of trainee success. In order to disentangle these factors, we developed a statistical model to measure the impact of each variables on the subsequent career of trainees. Our model considers two possible scenarios (Eq. 1): (a) the protégé moves on to a private sector position or to a public research position that does not involve training, thus excluding him from having descendants in the training network; and (b) the protégé moves on to an independent research position involving training, in which case their own proliferation rate is used to measure their human-capitalized scientific legacy. In order to permit adequate-time for measuring trainee proliferation over a 10-year window, we restricted our dataset to triplets where the protégé finished their latest training no later than 2007. All the variables were available at the completion of postdoctoral training and thus were not biased by the subsequent independent career of protégés. —

~~These outcomes are—~~ These scenarios were integrated into a mixed model ~~—Two model composition techniques are suitable to handle the large prevalence of postdocs without trainees, the hurdle and zero-inflated frameworks⁷. They differ by their modeling of researchers without trainees: the hurdle framework assumes that all independent researchers have at least one trainee, while the zero-inflated framework allows the existence of some independent researchers that have no trainee in the database. This latter scenario would correspond either to incomplete Academic Tree profiles or to researchers not involved in graduate/postdoctoral training. Furthermore, that predicted~~ the proliferation rate ~~may be modeled as a Poisson distribution or as a negative binomial distribution. The former assumes that count variance is directly proportional to mean count, while the latter relaxes this assumption and allows over-dispersion, at the cost of an extra free parameter.~~

~~To decide which of the four model architectures of trainees based on variables characterizing their mentor relationships. We simultaneously determined the best model architecture (hurdle or zero-inflated with Poisson or negative binomial distribution) and which predictors yielded the best fit to the data, we screened the performance of each architecture on all possible combinations of predictors. This was done by calculating parameter values for each model that maximized, see Methods) and the best combination of variables through a cross-validated search conducted over all predictor combinations. The best-fitting model overall was the zero-inflated negative binomial model (Supplementary Fig. 16). We focus on this model for the remainder of the paper. Quantitative details of the model comparison, including the cross-validated log-likelihood of predicted outcomes, using a 10-fold cross-validation scheme. Shapley values were then computed scoring the relative contribution of each variable scores of alternative architectures, are available in Supplementary Information.~~

The relevance of individual variables was assessed using their Shapley scores⁵⁰, which indicate their relative contribution to the overall model’s performance. Negative binomial distributed count rates consistently outperformed count rates conforming to a Poisson distribution, indicating the presence of over-dispersion in rates for the highest training researchers. Zero-inflated models performed slightly better than hurdle models goodness of fit (Table 1), and using their ranks from the Forward- and Backward Selection Algorithms

	Goodness-of-fit	Variable ranks		
		SV	FSA	BSA
Temporal trend				
Training end year	1.822	1	1	2
Postdoc duration	-0.027	10	11	11
Network				
Graduate mentor proliferation	1.301	3	3	3
Postdoc mentor proliferation	1.764	2	2	1
Mentor graph distance	-0.109	12	12	12
Graduate mentor age	-0.010	8	7	8
Postdoc mentor age	0.121	6	6	7
Publication				
Mentors publication similarity	0.369	4	4	5
Graduate mentor/trainee similarity	-0.025	9	10	10
Postdoc mentor/trainee similarity	0.163	5	5	4
Publications with graduate mentor	0.090	11	9	6
Publications with postdoc mentor	-0.004	7	8	9

Table 1: Overview of factors impacting trainee proliferation and their contribution to the model’s goodness-of-fit. Highlighted variables ~~are the ones were~~ kept in the model. The contribution of each variable to the overall ~~fitness model performance~~ is ~~its~~ Shapley value, with positive values denoting ~~better fit~~ a ~~greater contribution~~. The 12 variables are ranked according to their increasing order of importance, according to ~~the rank of~~ their Shapley Value (“SV”), the ~~rank when using the~~ Forward Selection Algorithm (“FSA”) and the ~~rank when using the~~ Backward Selection Algorithm (“BSA”).

~~(FSA and BVA, see Methods). Several variables had a negative Shapley score, indicating that an assumption~~
~~that all permanent research faculty must have at least one trainee is not consistent with this dataset (they~~
~~tend to form spurious relationships with the protégé’s proliferation. In particular, the mentor network~~
~~distance, academic age of graduate mentor, and postdoctoral training duration had negative Shapley scores.~~
~~These same variables were also excluded by the iterative variable selection (both FSA and BSA, Ta-~~
~~ble 31). The best-fitting model was overall the zero-inflated negative binomial model, whose cross-validated~~
~~predictions are shown along each dimension in Fig. 16 in Supplementary Information. We focus on this~~
~~specific mathematical model for the remainder of the paper. Thus, we excluded them from further analysis. The~~
~~relevance of the number of publications with the postdoc mentor was more ambiguous: it had a negative~~
~~Shapley value but was ranked above the number of publications with graduate mentor by the forward CSA~~
~~algorithm. Also, this metric is widely used to evaluate job applicants and has been reported as the most~~
~~important metric used by search committees, above the quality of journals or the funding track-record⁵³.~~
~~Thus we opted to include it in subsequent analyses.~~

~~Several variables had a negative Shapley score (Table 1), indicating model over-fitting. These variables~~
~~tend to form spurious relationships with the protégé’s proliferation, limiting their relevance for modeling. In~~

particular, three variables exhibited high levels of over-fitting: mentor network distance, the academic age of
graduate mentor, and the total time spent by the protégé in postdoctoral studies. These same variables were
also systematically excluded by the iterative variable selection processes based on Shapley values (both CSA
forward and backward algorithms, Table 1), thus we excluded them from the modeling. The relevance of the
number of publications number with postdoc mentor is more ambiguous: it has a negative Shapley value,
but is ranked above the number of publications with graduate mentor in the forward CSA algorithm. This
metric is widely used to evaluate job applicants (for example, it has been reported as the most important
metric used by committees studied in ⁵³, ranking above the quality of journals or the funding track record);
thus we opted to include it in the model.

2.4 Determinants of academic success

The impact of mentorship variables on the odds of trainees obtaining of an independent research position
and on their long-term training proliferation rates are summarized in Table 1 and Fig. 4. The odds of
continuing in academia were positively influenced by higher mentor proliferation rates, greater postdoctoral
mentor academic age, and close publication similarity between the protégé and their postdoctoral men-
tor (Figs. 4 and 16B,C). In contrast, training end year and high mentor publication similarity negatively
influenced the probability of continuing in academia (Figs. 4 and 16A,D).

A similar, but not entirely overlapping set of variables influenced the protégé's long-term proliferation
rate. Trainee proliferation was positively influenced by the mentors' proliferation rates, along with semantic
proximity to the postdoctoral mentor and the number of publications co-authored with the graduate mentor
(Figs. 4 and 16G-I). Overall, these results show that highly prolific mentors tend to provide their protégés
with the assets required for their own success in academia, both in terms of securing permanent research
positions and of increasing their long-term proliferation.

Figure 4: Modeled effects of variables for continuing in academia (left column) and for training rate when continuing (right column). The probabilities depict the change in the odds of continuing in academia (left) and in the training rate (right) induced by an increase of one unit for each variable. The error bars show the 95% bootstrapped confidence interval. Width of the bars and error bars show the relative importance (z-scores, see scale bar) whereas the percentages show the effect of adding one original unit. Significant changes are color-coded and the associated effects are shown in bold. For example, the interpretation for the “Postdoc mentor rate” variable is as follow: *ceteris paribus*, an increase of 1-point on the postdoc mentor proliferation scale (i.e. one more trainee per decade) results in increased odds by **9.38.9%** of the protégé to find a permanent position, and this effect is statistically significant. The long term effect this change is also significant, and the protégé’s proliferation rate is then increased by 2.6%.

9 ↑

The effect of training end date on the odds of continuing in academia was found to be very strong
 (Figs. 4 and 16A), consistent with known long-term trends toward a shortage-decreasing number of indepen-
 dent academic positions available to postdoctoral trainees (e.g. 39,41? 28,39,41). We considered the possibility
 that ~~temporal bias in the~~ collection-of-some-sampling-of-other variables could confound their effects with
 this strong temporal trend. To control for this possibility, we fit the same models using a temporal subset of
 data and without ~~using-the~~ training end date ~~to confirm the robustness of non-temporal variables~~ (Fig. 11
 in Supplementary Materials). We also evaluated alternative temporal models, which included either a
 polynomial expansion of training end date or training end date as an ordinal variable coding for different
 temporal epochs (Figs. 12 and 13 in Supplementary Information). These control models reveal the same
 effects for the non-temporal variables, confirming that the significant effects in Fig. 4 are not confounded by
 temporal trends.

Features-Characteristics of the postdoctoral mentor generally had greater influence on trainee success
 than the graduate mentor. This suggests a dominant role of postdoctoral advisors-mentors on the future
 career of protégés. The age of the postdoctoral mentor contributed to the likelihood of the protégé securing

10 ↑

a permanent position, ~~with~~ , with increased odds for older postdoctoral mentors ~~improving the odds~~. In
 contrast, graduate mentor age did not have a significant influence. We tested whether a possible influence
 of the graduate mentor’s seniority was masked by correlations with features of the postdoctoral mentor. We
 fit an alternative, partial model based only on features of the graduate training. This model ruled out the
 possibility of confounds with the postdoctoral mentor, as it produced in coefficients similar to the full model,
 again revealing no influence of graduate mentor academic age (Fig. 8 in Supplementary Materials).

Network variables were broadly found to ~~have~~ make a larger contribution to overall model performance
 than publication variables (Fig. 5). This relatively greater influence is consistent with the Shapley values
 and the variable ordering in Table 1, and is found across all formulations of the model studied (Table 3).

Figure 5: Impact of adding the different groups of variables on the log-likelihood of the model. Once the temporal trends are accounted for, the largest effect corresponds to the inclusion of mentor network features (training rate and academic age).

We restricted our main analysis to data that was available before the end of training ~~, so as~~ to avoid any
 confound associated with continuing versus not continuing in academia. To investigate whether the seman-
 tic content of papers published after the end of the postdoc continue to influence the career outcomes, we
 further included them as extra variables in the model. We observed ~~a large explanatory power of~~ substantial
 explanatory power for this late-publication similarity in explaining ~~the~~ continuation in academia, and spe-
 cially so for the postdoc ~~advisor-trainee~~ advisor-trainee similarity (supplemental Fig. 17). This ~~strongly~~
 finding suggests that strong ties formed during training and ~~evolving toward~~ transitioning into
 a collaboration with the former advisors has a beneficial impact on the trainee’s s career. This also reinforces the idea
 that the postdoctoral advisor has a larger influence on the future career than the graduate advisor, as was

11 ↑
12 ↑

found using variables available at the end of the postdoc (~~supplemental~~ Supplemental Figs. 4 and 17).

**2.5 Consistency of effects across fields**

To assess the consistency of the effects across fields, we split the data in two subsets: neuroscience only
and other life sciences (Fig. 7). These non-overlapping datasets show similar, albeit noisier, properties
than the full dataset reported in main text (Fig. 2 vs. Figs. 14 and 15 in Supplementary Information).
The mentorship patterns of Fig. 2A-D are comparable across subsets. Both also show the same trends
of increasing postdoctoral trainee numbers and training duration (Fig. 2E-F) and the same patterns of
publication similarity (Fig. 2G-H). Importantly, models computed for both data subsets showed the positive
effect of intellectual synthesis, with a strong effect of mentor publication similarity in both cases. Thus, the
advantage of trainees performing intellectual synthesis generalizes across the life sciences. We do observe some
discrepancies between the data subsets. Fewer variables achieve significance in the non-neuroscience subset,
particularly for the long-term prediction of trainee proliferation rate, possibly reflecting the smaller sample
size. In addition, the proliferation rate of the graduate mentor shows a substantially stronger influence in the
non-neuroscience subset, while the effect of postdoctoral advisor proliferation rate fails to reach significance.
Whether this reflects a differential impact of the postdoctoral advisor between neuroscience and other life
sciences, or whether this is an artifact of limited sampling remains to be investigated.

13 ↑

276 **2.6 Nonlinear influence of mentor graph distance**

[revised manuscript text omitted]

3 Discussion

We identified ~~a set of~~ factors related to mentorship that influence the academic success of postdoctoral trainees
in biomedical research. We considered two measures of ~~academic success: obtaining success: whether or not~~
~~a trainee obtains~~ an independent research position and ~~the their proliferation rate~~ (number of researchers
trained(~~proliferation rate~~) once a position is obtained. The ~~main~~ factors influencing the likelihood of ~~trainees~~
obtaining an independent ~~research~~ position were (a) the ~~date of entry in year of entry to~~ the job market
(~~reflecting long-term trends in job openings~~), (b) the success of their mentors (i.e., ~~their mentor~~ proliferation
rate), (c) ~~the ability of the trainee their ability~~ to synthesize between the intellectual output of their mentors,
and ~~lastly~~-(d) the professional age (time since graduation) of ~~their the~~ postdoctoral mentor. The main ~~drivers~~
~~of the trainee’s proliferation, after obtaining an independent position, predictors of trainee proliferation~~
were the mentors’ training rates and publishing research that was similar to that of their graduate mentor.
~~Postdoctoral training duration and Neither the duration of training nor~~ the professional age of the graduate
mentor ~~were found to be irrelevant to trainees impacted trainee~~ success. Below, we discuss these findings
and their relevance to the existing literature.

**Trainee synthesis.** This study reveals the importance of intellectual synthesis in the pursuit of an inde-
pendent research career in life sciences. Trainees tended to be more successful if publications by their grad-
uate and postdoctoral ~~had weak mentors had low~~ semantic similarity, suggesting that their work links ideas
and/or approaches from two previously disparate subfields. This finding can be framed within the weak-ties
theory of Granovetter²⁵, which emphasizes the importance of bridges between ~~interconnected communities.~~
~~Applied to academic research, this theory posits some advantages for trainees of mentors belonging to different~~
~~scientific communities. In their early careers, trainees of disparate mentors benefit from weakly connected~~
~~communities. Trainees with disparate mentors may also benefit from more professional opportunities in~~
the larger combined network of their advisors, ~~which may be instrumental in creating more professional~~
~~opportunities and thus helping to secure a permanent position~~^{24,25}. In addition, researchers bridging dis-
disparate scientific communities are in a position to diffuse their innovations ~~more effectively, allowing them~~
~~to reach more peers and across a larger group of peers,~~ possibly garnering more recognition ~~in their late~~
~~careers for their work~~⁴⁹.

For these beneficial effects to ~~arise occur~~, the trainee’s research must ~~be able to act as a bridge actually~~
~~bridge the disparate training fields~~. Thus, it is important that the work maintains some similarity with that
of ~~both~~ the graduate and postdoctoral mentors. Indeed, ~~successful~~ trainees tended to have strong semantic

similarity to both ~~their graduate and postdoctoral~~ mentors, consistent with a meaningful intellectual impact
by both ~~mentors~~ on the trainee's work. This effect persists and is amplified when considering publications
~~made~~ after the completion of postdoctoral training, suggesting a long-lasting ~~benefice of trainees~~ benefit of
synthesis.

~~Given that having~~ Given that trainees benefit from mentors with dissimilar research ~~positively impacts~~
~~trainee careers~~, one might also expect ~~a large mentor network distance to also have a positive influence~~ trainees
to benefit from mentors separated by a large distance in the genealogy graph. However, mentor network
distance (*i.e.*, distance to a common ancestor) does not predict trainee continuation or proliferation (Table 1,
Fig. 6B). This ~~may reflect the fact suggests~~ that mentor graph distance ~~is~~ may be too crude a measure of
intellectual similarity to provide significant predictive power. Alternatively, ~~there may be a competing social~~
~~factor, where different social factors may influence the career path of~~ trainees with closely related mentors may
~~follow a different career trajectory than other trainees~~. The bimodal distribution ~~in the histogram~~ of mentor
graph distance suggests that there are in fact two groups of trainees. ~~Moreover, the~~ The decreased odds of
having ~~a scientific progeny when co-mentors are linked by three or less mentoring steps supports a hypothesis~~
~~that the associated lack of diversity could be detrimental to success~~. ~~Investigation of these fine patterns of~~
~~close relationships in training deserve attention in future work~~.

scientific progeny for the group with mentor network distance less than four is consistent with the
hypothesis that a lack of intellectual or social diversity is detrimental to professional success.

**Influence of mentors' success.** ~~Another insight of our study is to show~~ We also found that mentors'
~~training rate positively influences proliferation rates positively influence~~ the trainee's ~~training rate~~ proliferation
rate, consistent with previous findings ^{8,15,36}. The link between ~~mentors' and trainees' mentor and trainee~~
success in academia has been ~~reported previously~~ observed using other measures. In particular, ~~the mentor~~
~~'s mentor~~ prestige has been shown to be correlated with the trainee ~~'s~~ publication rate¹⁴. ~~Mentors' research~~
~~productivity has also been shown to have a direct impact on~~, and a mentor's research productivity impacts
both the prestige of ~~the first professional appointment of the trainee, and on research productivity at later~~
~~stage of their careers~~ a trainee's first professional research position and research productivity later in their
career³⁴.

~~This studies reveals a novel finding~~ For our bioscience data set, we also found that the proliferation of the
postdoctoral mentor has a greater effect than that of the graduate mentor on the odds of securing a permanent
position (Fig. 4). This ~~trend observation~~ is consistent with ~~the study of~~ Long et al.³⁵, which showed that the
prestige of the postdoctoral institution has a stronger positive effect than that of the doctoral institution
for a cohort of biochemists graduated between 1957 and 1963. The greater influence of the postdoctoral
advisor is also consistent with the idea that the postdoc is a launching pad for an independent career, ~~where~~
during which time scientists build up critical secondary skills (~~be it~~ network connections, publication and
grant funding track-record, technical expertise) required for ~~obtaining a permanent research position~~.

an independent position. Our analysis also suggests that the benefits of training with a successful mentor
and pursuing research along similar intellectual lines may be related, as successful mentors are likely to
work on topics of broad relevance to their field. Overall, our study ~~justifies~~ supports the long-standing
advice that prospective students should look at the training ~~track-record~~ record of potential mentors to assess
their quality^{5,26}. ~~We show here that besides boosting the odds of securing a permanent research position,~~
~~highly prolific mentors also tend to have highly prolific trainees, a desirable quality as it is globally linked~~

~~to academic achievements~~ ^{8,15?} ~~.~~

**Postdoc duration.** ~~Although it~~ It is sometimes thought that ~~postdoctoral training duration~~ the duration
of postdoctoral training has increased in recent decades, ~~quantitative reports have consistently.~~ Quantitative
reports have showed a stabilization (e.g. ²⁸). ~~This~~ However, most previous work draws on data from the
~~Survey of Doctorate Recipients (SDR), which is limited to US-graduated researchers. Thus it~~ ~~discount~~
~~trends~~ omits data for international postdocs, who often remain longer in postdoctoral positions due to visa
limitations³¹. In the current study, ~~We report that the total~~ we find that the mean duration of postdoctoral
training in life ~~science~~ sciences has increased, from less than 3 years in 1970 to 4.1 years in 2015 (Fig. 2F).
~~We also report that the proportion of long~~ The proportion of long-term postdocs (> 8 years) has remained
stable at around 10%. The absence of an increasing trend in long-term positions may be seen as encouraging.
~~Yet, the situations of the 10%~~ However, the persistently large number of “permadoes” remains worrisome
and fits in the narrative that sometimes ~~postdocs~~ are more a source of cheap labor than a meaningful step
on the pathway of career development²¹.

Previous work on the influence of postdoctoral training ~~time~~ duration on the odds of securing a permanent
position show ~~opposite results.~~ inconsistent results. Yang and Webber⁶¹ recently showed that ~~earning~~
completing 2 to 4 postdoctoral appointments nearly doubled the odds of obtaining an academic position,
~~but although it~~ did not enhance ~~the~~ long-term productivity ~~of researchers~~. In a study on biochemists, Nerad
and Cerny³⁹ showed, on the contrary, that relatively short postdocs (< 5 years) led to better ~~career outcomes~~
in academia outcomes in academic careers. The ~~objective benefits of completing several postdocs~~ beneficial
experience of extended postdoctoral training must also be put in balance with a form of survivor’s bias,
where those that can afford the cost of long-term postdocs tend to exhibit characteristics correlated with the
odds of securing a permanent ~~positions~~ position⁶¹. ~~In particular, long-term postdoctoral trainees fit more~~
~~closely to the profiles~~ Long-term postdoctoral trainees typically fit the profile of tenured researchers, as they
tend to originate less often from under-represented minorities and are more often ~~males~~ ^{10,53} male ^{33,53}. The
impact of long-term postdoctoral training is then hard to assess in isolation (e.g., women ~~’s more frequent~~
~~departure from science~~ more frequently depart from science after a first postdoc to take care of children ~~is~~
~~not linked to scientific ability and rather finds its root in~~ because of traditional family structures, cf. Chen
et al.¹⁰, Ginther and Kahn²³, Martinez et al.³⁷). ~~Our study contributes to this ongoing debate by showing~~
Our study finds no systematic link between training duration and academic success. ~~Yet, our study can~~
~~not rule out the possibility of a spurious negative effect of long cumulative postdoc duration that is masked~~
~~by independent variables~~ However, we can not exclude a potential negative influence of long postdocs that
might be masked by confounds like gender or minority status, as these variables were not included in the
model.

**Mentor experience.** The academic age of the postdoctoral mentor at the time of training had a
~~weak~~ small but significant positive impact on ~~academic continuation~~ the odds of continuing in academia
(Figs. 4 and 16F). ~~On the contrary~~ In contrast, graduate mentor age was ~~not informative of academic~~
~~success.~~ not predictive. Several factors might explain the ~~positive impact of more experienced mentors:~~
greater experience benefit of a more experienced mentor: greater practical knowledge, more material resources
in a stable, well-funded laboratory, and better social connections in the network likely to hire the trainee ~~for~~
~~an independent position~~. Future studies controlling specifically for these factors ~~are required to determine~~
may elucidate their relative importance.

Of interest, the study of Malmgren et al.³⁶ reported opposite effects ~~in~~ for the field of mathematics;
~~where~~. For mathematicians, the age of the graduate mentor was negatively ~~linked to~~ correlated with trainee
proliferation. ~~In our study of biosciences, we found a positive effect of postdoctoral mentor academic age on~~
~~securing a permanent position; and no effect of graduate mentor age on either continuation or proliferation.~~
~~This discrepancy~~ This difference from the positive effect we observe for mentor age in life sciences may arise
from the different populations of researchers ~~sampled~~. This study considered scientists trained ~~by both~~
~~a graduate and postdoctoral mentor~~ in recent decades, whereas Malmgren et al.³⁶ limit their analyses to
Ph.D. students graduating between 1900 and 1960. Our study also focused on researchers trained by both
a graduate and postdoctoral mentor rather than dyads formed by just a graduate advisor and student.
~~In addition~~, Finally, the academic opportunities of mathematics graduates are generally thought to be
more advantageous than in biosciences^{4,39}, making direct comparison between the two datasets difficult.
Regardless of the cause, these differences suggest that the features of good mentors vary depending on the
broader social and academic context. It may be that effective mentoring strategies and profiles depend on
the job market faced by trainees.

[revised manuscript text omitted]

5.2 Selection criteria.

We identified 20,695 triplets in the Academic Tree database, each consisting of a trainee with one graduate and one postdoctoral mentor. In some outlier cases, more than four graduate or postdoctoral advisors were listed, resulting in the same protégé contributing to many triplets. To avoid over-counting these trainees, we removed data for trainees with more than four graduate or postdoctoral mentors. This resulted in 18,856 triplets encompassing 12,853 unique trainees, 9,111 unique graduate mentors and 7,322 unique postdoc mentors.

5.3 Missing date inference.

Start and end dates of training were entered optionally by users through the web interface. In 49% of the triplets, both start and end dates were available. Training end dates were more relevant to the analyses presented here, as they marked the transition to the status of independent researcher. When both dates were missing, we inferred the end date by identifying the earliest commonly authored publication and adding the field-specific median lag between the start year and first publication. This was performed separately for graduate and postdoctoral training, based on local regression models (LOESS,¹²) to account for changes in training duration over time. For trainees without an end date but with a start date, we estimated the missing date information by adding the mean difference computed from trainees with complete data, again adjusting for changes in the duration of training periods over time. Using this approach, we were able to

Figure 7: Statistical mode of the mentor proliferation rates (horizontal axis) and sample size (number of triplets along the vertical vertical axis). Life Sciences (red) are well represented in the Academic Family Tree dataset, and thus were the focus of the current study. The In most sub-fields, the annual proliferation rate is between 0.05 (one trainee every 20 years) 0.5 and 0.1-1 (one new trainee every 10 years year or every other year) in most of its sub-fields. Maths and Engineering fields (green) show generally higher proliferation rates, while Physical Sciences (blue) and Social Sciences (cyan) show lower rates.

520 assign end dates for training period for 90% of the graduate and 89% of the postdoctoral relationships (see
 521 Supplementary Table 2). Overall, 15,583 triplets (83% of the triplets) had dates that could be fully inferred,
 522 both at the graduate and postdoctoral levels.

5.4 Semantic analysis.

Academic Family Tree researchers were linked to publications in the Medline and Scopus databases using a simple disambiguation procedure. ~~Publications for which an author name matched a researcher name were attributed to that researcher with high (likely match) or low probability (unlikely) based on~~ 
[revised manuscript text omitted]

- 2. Akil, H., Balice-Gordon, R., Cardozo, D. L., Koroshetz, W., Norris, S. M. P., Sherer, T., Sherman, S. M.
and Thiels, E. 2016, ‘Neuroscience training for the 21st century’, *Neuron* **90**(5), 917–926.
- 3. Aumann, R. J. 1989, Game theory, *in* ‘Game Theory’, Springer, pp. 1–53.
- 4. Austin, J. 2013, ‘~~Want to Be a Professor? Choose Math,~~
~~url=http://dx.doi.org/10.1126/science.caredit.a1300150,~~ ~~DOI=10.1126/science.caredit.a1300150,~~
~~journal=Science~~Want to be a professor? choose math’.
- 5. Barres, B. A. 2013, ‘How to Pick a Graduate Advisor’, *Neuron* **80**(2), 275 – 279.
- 6. Boadi, K. 2014, ‘Erosion of funding for the national institutes of health threatens us leadership in
biomedical research’, *Center for American Progress* **25**.
- 7. Cameron, A. C. and Trivedi, P. K. 2013, *Regression analysis of count data*, Vol. 53, Cambridge university
press.
- 8. Chariker, J. H., Zhang, Y., Pani, J. R. and Rouchka, E. C. 2016, ‘Identification of Successful Mentoring
Communities using Network-based Analysis of Mentor-Mentee Relationships across Nobel Laureates’,
*bioRxiv* p. 075432.
- 9. Check, H. E. 2016, ‘Young scientists ditch postdocs for biotech start-ups.’, *Nature* **539**(7627), 14.
- 10. Chen, J., Kim, M. and Liu, Q. 2016, ‘Do Female Professors Survive the 19th-Century Tenure System?:
Evidence from the Economics Ph. D. Class of 2008’.
- 11. Chevan, A. and Sutherland, M. 1991, ‘Hierarchical partitioning’, *The American Statistician* **45**(2), 90–
96.
- 12. Cleveland, W. S., Grosse, E. and Shyu, W. M. 1992, ‘Local regression models’, *Statistical models in S*
**2**, 309–376.
- 13. Cohen, S., Dror, G. and Ruppin, E. 2007, ‘Feature selection via coalitional game theory’, *Neural Com-
putation* **19**(7), 1939–1961.
- 14. Crane, D. 1965, ‘Scientists at major and minor universities: A study of productivity and recognition’,
*American sociological review* pp. 699–714.
- 15. Crosta, P. M. and Packman, I. G. 2005, ‘Faculty productivity in supervising doctoral students’ disserta-
tions at Cornell University’, *Economics of Education Review* **24**(1), 55–65.
- 16. David, S. V. and Hayden, B. Y. 2012, ‘Neurotree: A collaborative, graphical database of the academic
genealogy of neuroscience’, *PloS one* **7**(10), e46608.

- 17. Deerwester, S., Dumais, S. T., Furnas, G. W., Landauer, T. K. and Harshman, R. 1990, 'Indexing by
latent semantic analysis', *Journal of the American society for information science* **41**(6), 391.
- 18. Dutt, K., Pfaff, D. L., Bernstein, A. F., Dillard, J. S. and Block, C. J. 2016, 'Gender differences in
recommendation letters for postdoctoral fellowships in geoscience', *Nature Geoscience* **9**(11), 805–808.
- 19. Efron, B. 1979, 'Bootstrap methods: another look at the jackknife', *The annals of Statistics* pp. 1–26.
- 20. Fortunato, S., Bergstrom, C. T., Börner, K., Evans, J. A., Helbing, D., Milojević, S., Petersen, A. M.,
Radicchi, F., Sinatra, R., Uzzi, B. et al. 2018, 'Science of science', *Science* **359**(6379), eaao0185.
- 21. Freeman, R. B. 2002, 'Thanks for the great postdoc bargain', *Science's Next Wave* .
- 22. Gelman, A., Hwang, J. and Vehtari, A. 2014, 'Understanding predictive information criteria for Bayesian
models', *Statistics and Computing* **24**(6), 997–1016.
- 23. Ginther, D. K. and Kahn, S. 2006, 'Women's careers in academic social science: Progress, pitfalls, and
plateaus', *The Economics of Economists, A. Lanteri and J. Vromen, eds.(Cambridge, UK: Cambridge
University Press, 2014)* .
- 24. Granovetter, M. 1995, *Getting a job: A study of contacts and careers*, University of Chicago Press.
- 25. Granovetter, M. S. 1973, 'The strength of weak ties', *American journal of sociology* **78**(6), 1360–1380.
- 26. Hall, R. M. and Sandler, B. R. 1983, 'Academic mentoring for women students and faculty: A new look
at an old way to get ahead.'
- 27. Hofmann, T. 1999, Probabilistic latent semantic indexing, in 'Proceedings of the 22nd annual interna-
tional ACM SIGIR conference on Research and development in information retrieval', ACM, pp. 50–57.
~~Kahn, S. Ginther, D. K. 2017a, 'The impact of postdoctoral training on early careers in biomedicine',
*Nature Biotechnology* **35**(1), 90–94.~~
- 28. Kahn, S. and Ginther, D. K. 2017b, 'The impact of postdoctoral training on early careers in biomedicine',
*Nature Biotechnology* **35**(1), 90–94.
- 29. Kram, K. E. 1988, *Mentoring at work: Developmental relationships in organizational life.*, University
Press of America.
- 30. Kruskal, W. 1987, 'Relative importance by averaging over orderings', *The American Statistician* **41**(1), 6–
10.
- 31. Lan, X. 2012, 'Permanent visas and temporary jobs: evidence from postdoctoral participation of foreign
PhDs in the United States', *Journal of Policy Analysis and Management* **31**(3), 623–640.
- 32. Lane, J. and Bertuzzi, S. n.d., *The star metrics project: current and future uses for s&e workforce data.*

- 33. Layton, R. L., Brandt, P. D., Freeman, A. M., Harrell, J. R., Hall, J. D. and Sinche, M. 2016, 'Diversity
exiting the academy: Influential factors for the career choice of well-represented and underrepresented
minority scientists', *CBE-Life Sciences Education* **15**(3), ar41.
- 34. Long, J. and McGinnis, R. 1985, 'The effects of the mentor on the academic career', *Scientometrics*
**7**(3-6), 255–280.
- 35. Long, J. S., Allison, P. D. and McGinnis, R. 1979, 'Entrance into the academic career', *American*
*sociological review* pp. 816–830.
- 36. Malmgren, R. D., Ottino, J. M. and Amaral, L. A. N. 2010, 'The role of mentorship in protégé perfor-
mance', *Nature* **465**(7298), 622–626.
~~Malmgren, R., Ottino, J., Nunes, A. L. 2010, 'The role of mentorship in protégé performance.',~~
~~Nature465, 622–6.~~
- 37. Martinez, E. D., Botos, J., Dohoney, K. M., Geiman, T. M., Kolla, S. S., Olivera, A., Qiu, Y., Rayasam,
G. V., Stavreva, D. A. and Cohen-Fix, O. 2007, 'Falling off the academic bandwagon', *EMBO reports*
**8**(11), 977–981.
- 38. NAS 2014, 'National Academy of Sciences, National Academy of Engineering, Institute of Medicine. The
Postdoctoral Experience Revisited', *National Academies Press* .
- 39. Nerad, M. and Cerny, J. 1999, 'Postdoctoral Patterns, Career Advancement, and Problems', *Science*
**285**(5433), 1533–1535.
- 40. Nihm, S. D. 1976, 'Polynomial law of sensation.', *American Psychologist* **31**(11), 808.
- 41. NRC 1981, *National Research Council (United States), Committee on a Study of Postdoctorals in Science*
*and Engineering in the United States and National Research Council (US), Commission on Human*
*Resources. Postdoctoral Appointments and Disappointments: A Report of the Committee on a Study of*
*Postdoctorals in Science and Engineering in the United States*, number 3132, National Academy Press.
- 42. NSB 2016, 'National Science Board: Science and engineering indicators 2016', *NS Foundation (Ed.)*.
*Arlington, VA: National Science Foundation* .
- 43. Pan, R. K., Petersen, A. M., Pammolli, F. and Fortunato, S. 2016, 'The memory of science: Inflation,
myopia, and the knowledge network', *arXiv preprint arXiv:1607.05606* .
- 44. Pedregosa, F., Varoquaux, G., Gramfort, A., Michel, V., Thirion, B., Grisel, O., Blondel, M., Pretten-
hofer, P., Weiss, R., Dubourg, V. et al. 2011, 'Scikit-learn: Machine learning in Python', *Journal of*
*Machine Learning Research* **12**(Oct), 2825–2830.
- 45. Petersen, A. M., Fortunato, S., Pan, R. K., Kaski, K., Penner, O., Rungi, A., Riccaboni, M., Stanley,
H. E. and Pammolli, F. 2014, 'Reputation and impact in academic careers', *Proceedings of the National*
Academy of Sciences **111**(43), 15316–15321.
- 46. Petersen, A. M., Riccaboni, M., Stanley, H. E. and Pammolli, F. 2012, 'Persistence and uncertainty in
the academic career', *Proceedings of the National Academy of Sciences* **109**(14), 5213–5218.

- 47. Powell, K. 2015, ‘The future of the postdoc’, *Nature* **520**(7546), 144.
- 48. Ragins, B. R., Kram, K. E., Ragins, B. and Kram, K. 2007, ‘The roots and meaning of mentoring’, *The*
*handbook of mentoring at work: Theory, research, and practice* pp. 3–15.
- 49. Rogers, E. M. 2010, Diffusion of innovations, Simon and Schuster, chapter 8.
- 50. Shapley, L. 1953, ‘A value for n-person games’, *Contributions to the Theory of Games (Edited by H. W.*
*Kuhn and A. W. Tuck* **2**, 307–317.
- 51. Sinatra, R., Wang, D., Deville, P., Song, C. and Barabási, A.-L. 2016, ‘Quantifying the evolution of
individual scientific impact’, *Science* **354**(6312), aaf5239.
- 52. Smyth, P. 2000, ‘Model selection for probabilistic clustering using cross-validated likelihood’, *Statistics*
*and computing* **10**(1), 63–72.
- 53. Steinpreis, R. E., Anders, K. A. and Ritzke, D. 1999, ‘The impact of gender on the review of the curricula
vitae of job applicants and tenure candidates: A national empirical study’, *Sex roles* **41**(7), 509–528.
- 54. Stone, M. 1977, ‘An asymptotic equivalence of choice of model by cross-validation and Akaike’s criterion’,
*Journal of the Royal Statistical Society. Series B (Methodological)* pp. 44–47.
- 55. Stufken, J. 1992, ‘Letters to the Editor: On Hierarchical Partitioning’, *The American Statistician*
**46**(1), 70–77.
- 56. Su, X. 2013, ‘The impacts of postdoctoral training on scientists’ academic employment’, *The Journal of*
*Higher Education* **84**(2), 239–265.
- 57. Sugimoto, C. R. and Cronin, B. 2012, ‘Biobibliometric profiling: An examination of multifaceted
approaches to scholarship’, *Journal of the American Society for Information Science and Technology*
63(3), 450–468.
- 58. Trix, F. and Psenka, C. 2003, ‘Exploring the color of glass: Letters of recommendation for female and
male medical faculty’, *Discourse & Society* **14**(2), 191–220.
- 59. Waltman, L. 2016, ‘A review of the literature on citation impact indicators’, *Journal of Informetrics*
10(2), 365–391.
- 60. Way, S. F., Morgan, A. C., Clauset, A. and Larremore, D. B. 2017, ‘The misleading narrative of the
canonical faculty productivity trajectory’, *Proceedings of the National Academy of Sciences* p. 201702121.
- 61. Yang, L. and Webber, K. L. 2015, ‘A decade beyond the doctorate: the influence of a US postdoctoral
appointment on faculty career, productivity, and salary’, *Higher Education* **70**(4), 667–687.
- 62. Yin, X., Han, J. and Philip, S. Y. 2007, Object distinction: Distinguishing objects with identical names,
in ‘Data Engineering, 2007. ICDE 2007. IEEE 23rd International Conference on’, IEEE, pp. 1242–1246.
- 63. Zeileis, A., Kleiber, C. and Jackman, S. 2008, ‘Regression models for count data in R’, *Journal of*
*statistical software* **27**(8), 1–25.

Supplementary information

1

2 Availability of date information

Availability of date information	
Start of PhD	51%
End of PhD	66%
Earliest publication date with graduate advisor	74%
At least one graduate date available	90%
Start of Postdoc	57%
End of Postdoc	39%
Earliest publication date with graduate advisor	71%
At least one postdoc date available	89%

Table 2: Statistics of data availability based on publications and manually entered dates

6 Model and variable selection

Two model composition techniques are suitable to handle the large prevalence of postdocs without trainees, the *hurdle* and *zero-inflated* frameworks⁷. They differ by their modeling of researchers without trainees: the hurdle framework assumes that all independent researchers have at least one trainee, while the zero-inflated framework allows the existence of some independent researchers that have no trainee in the database. This latter scenario would correspond either to incomplete Academic Tree profiles or to researchers not involved in graduate/postdoctoral training. Furthermore, the proliferation rate may be modeled as a Poisson distribution or as a negative binomial distribution. The former assumes that count variance is directly proportional to mean count, while the latter relaxes this assumption and allows over-dispersion, at the cost of an extra free parameter.

To decide which of the four model architectures (hurdle or zero-inflated with Poisson or negative binomial distribution) and which predictors yielded the best fit to the data, we screened the performance of each architecture on all possible combinations of predictors. This was done by calculating parameter values for each model that maximized log-likelihood of predicted outcomes, using a 10-fold cross-validation scheme. Shapley values were then computed scoring the relative contribution of each variable to the overall model's performance. Negative binomial distributed count rates consistently outperformed count rates conforming to a Poisson distribution, indicating the presence of over-dispersion in rates for researchers with the highest proliferation rate. Zero-inflated models performed better than hurdle models, indicating that an assumption that all independent research faculty have at least one trainee is not consistent with this dataset (Table 3). The best-fitting model overall was the zero-inflated negative binomial model. Cross-validated predictions for each input variable are shown in Fig. 16 in Supplementary Information. Given its superior performance, we focus on this mathematical model for the main results of the paper.

	$\Delta\mathcal{L}$ in cross-validation			
	HP	ZIP	HNB	ZINB
temporal	-10.24	-10.23	-0.47	-0.45
network	-9.45	-9.40	-0.54	-0.30
publications	-10.42	-10.38	-0.71	-0.52
temporal + network	-9.12	-9.11	-0.07	-0.05
temporal + publications	-10.10	-10.09	-0.39	-0.38
network + publications	-9.31	-9.26	-0.47	-0.24
temporal + network + publications	-8.97	-8.96	-0.03	0.00

Table 3: Impact of model types and predictors on predictive accuracy. The mathematical models compared are: the hurdle Poisson model (HP), hurdle negative binomial models (HNB), zero-inflated Poisson models (ZIP) and zero-inflated negative binomial models (ZINB). Predictors were grouped under the same categories as in Table 1: “temporal” for the year of the end of the postdoctoral appointment as well as the total duration of postdoctoral training; “network” for the proliferation of mentors, their academic age at training and their common ancestor distance; and “publications” for the publication similarity between mentors (prior to meeting the trainee) and the publication similarity between postdoctoral trainee and mentor (prior to training). The values displayed are cross-validated log-likelihood aligned on the best model (“ZINB” model using “network + publications + temporal” variables), with higher values denoting more accurate models.

7 Influence of graduate mentor's academic age

Fig. 8 show the regression coefficients computed using only features from graduate studies (Fig. 8). We find similar coefficients in these partial regressions, showing that the effects of graduate mentor features are independent from the effects of postdoc mentor features. Of interest, graduate mentor age is still non-significant in this analysis, showing that the lack of impact of graduate mentor age is not caused by features from the postdoctoral mentor. As this extra analysis was computed on the same triplet dataset, it does not rule out the more general potential impact of features from the graduate advisor / trainee dyad, in encouraging/discouraging a PhD to continue to the postdoc step.

25 ↑

Figure 8: Coefficients of a model focused on graduate mentor features (academic age, training rate, co-publishing and similarity with trainee) and excluding features related to postdoctoral studies.

³³ **8 Alternative models of postdoctoral preponderance**

Figure 9: Alternative model computed with an additional interaction term, “Postdoc ÷ graduate mentor rates”, which was designed to be high for trainees that moved to a more prolific postdoctoral mentor (“upward mobility” hypothesis). The lack of significance of this additional term showed that there is no systematic benefit associated with such a strategy.

Figure 10: Alternative model computed with a new interaction term, “Postdoc mentor rate × similarity”, contrasted with a second term: “Graduate mentor rate × similarity”. The first term should be high for trainees who moved into the “better” subfield of their postdoc mentor, while the second term contrasts this effect and should be high for trainees who stayed in the “better” subfield of their graduate studies. These terms were not linked to increased (or decreased) odds of finding a permanent position in this dataset. Interestingly, there is a slight influence of the “Postdoc mentor rate × similarity” term on long-term proliferation. One interpretation is that it corresponds to a long-term fatigue effect of disengagement from opportunistic trainees who embraced the research line of their postdoctoral mentor. However, it may also be a spurious effect that parallels the increased long-term effect of the “Postdoc mentor rate” in this regression, compared to the original regression.

9 Time-dependence of regression coefficients

Fig. 11 shows the regression coefficients obtained when training the model on temporal subsets of the data and without the time-controlling variable of “postdoc end year”. Except for this omission, the variables included in the model were the ones obtained after the selection process (cf. Table 1 in main text). The optimized coefficients from Fig. 11 display much more variability than with the regression shown in main text, due to lower sample sizes and the exclusion of temporal variables, but overall display similar trends as the full model.

We also controlled for long term temporal effects through a series of higher-order terms into the regression model (Fig. 12). This expansion of higher-order terms has the advantage of modeling arbitrary temporal trends, as in Taylor series, to the risk of over-fitting temporal trends⁴⁰. Here, not surprisingly these additional terms were able to capture some additional variance in the data, but they had no large impact on the other factors of interest in the regression. In particular, the intellectual synthesis effects appeared robust to the inclusions of finer temporal controls.

Figure 11: Regression coefficients obtained when optimizing the model on different temporal subsets while removing the postdoc end date from the set of predictors. Temporal partitions were chosen to contain roughly equal sample sizes, and were as follows: 1980 – 1992 ($n = 1,436$); 1992 – 1998 ($n = 1,337$); 1998 – 2002 ($n = 1,347$); 2002 – 2005 ($n = 1,404$); 2005 – 2007 ($n = 1,343$). The coefficients estimated to predict continuation in academia (A) and long-term mentoring rate (B) are consistent with the ones obtained with the full model that includes temporal predictors (represented in these plots as shaded areas). Specifically, whenever a variable reaches significance within a temporal subset, it does so in the same direction as the trend apparent in the full model. Reciprocally, whenever a variable reaches significance in the full model, there is at least one temporal subset when it reaches significance as well.

Figure 12: Model coefficients, similar to Fig.4 in main text but with the inclusion of more complex temporal trends, captured here through a series of higher-order terms: (Training end year)², (Training end year)³ and (Training end year)⁴.

47 We also tried another type of non-linear modeling of temporal effects, by replacing the continuous time
48 variable (“Training end year”) by an ordinal variable capturing the same temporal subdivisions as in Fig.4
49 (“Training end epoch”, numbered from 1 to 5). This transformation did not affect which coefficients were
50 significant nor did it change the order of magnitude of their effects (Fig. 13).

Figure 13: Model coefficients, similar to Fig.4 in main text but with a simpler temporal modeling that uses the ordinal variable “Training end epoch”.

⁵¹ **10 Consistency of effects across fields**

⁵²

27 ↑

Figure 14: Same as Fig. 2 in main text, but using data from non-neuroscience graduates ($n = 3,271$ triplets).

Figure 15: Same as Fig. 2 in main text, but using data of neuroscience graduates (\$n = 14,953\$ triplets).

11 Cross-validated model predictions and data

Figure 16: Academic continuation in life science (panels A to F) and modeled training rate (panels G to I) explained by the most significant variables. For each variable, the data is shown in black, and the cross-validated predictions of the model including this variable is shown in orange (“full model”). To visualize the contribution of each variable, we also display the cross-validated predictions of the model without this variable in purple (“partial model”). Lines and shaded areas represent respectively the mean values and their 95% bootstrapped confidence interval.

Figure 17: Contribution of publication similarity variables to the odds of continuing an academic career. In each of these plots, the data is shown in black, and the cross-validated predictions of the model including this variable is shown in orange (“full model”). To visualize the contribution of each variable, we also display the cross-validated predictions of the model without this variable in purple (“partial model”). Lines and shaded areas represent respectively the mean values and their 95% bootstrapped confidence interval. Overall, proximity with the postdoctoral advisor has a larger influence than proximity with the graduate advisor. Also, the mentors/trainee proximity displays a larger explanatory power when computed on publication data after the end of training (compared to data available at the end of the postdoc, cf. supplemental Fig. 16), with higher similarity linked to better odds of continuing in research.

Reviewers' Comments:

Reviewer #1:

Remarks to the Author:

I commend the authors on all the work they did to improve the manuscript. I think the additional tests dramatically strengthen the confidence on the reported results.

However, I still think that several changes need to be made. First and foremost, I believe that the analyses in the main text should be restricted to neuroscience. The number of triplets for neuroscience is about six-fold the number for all other life science fields.

I think the main text could state that analyses for other life science disciplines are not in conflict with the results for neuroscience but I do believe that the appearance that all results hold for all life science fields is an over-reach.

Second, I believe that the analyses reported in Fig. 3 can be improved. The data in Fig 2A shows that the replication rates of mentors spreads over a range of values that are actually quite different: 0.5 trainees/decade means about two trainees over a 40 year career, 5 trainees/decade means 20 trainees over a 40 year career. But there is even the problem with some mentors likely having even higher replication rates. Instead of plotting the rather uninteresting distributions shown in Figs. 3A-C, I recommend partitioning the mentors into several groups according to their replication rates and showing box plots of values of the different quantities for the different groups (and the results of a test comparing the means for the difference groups, instead of using the KS test). For the case of Fig. 3D, I would plot the value of the mean and skewness versus mentor group.

Third, I realized that I am not sure what the bars and scale represent in Fig. 4 and similar figures. I think it would be more useful to have the bars identifying the 95% CI for the estimates of the coefficients.

Reviewer #2:

Remarks to the Author:

I am satisfied by the authors' revisions and comments in response to the remarks in my report and those of the other the other two reviewers. While there are limitations to the data set the authors have studied in this work, I believe it is robust enough to support their conclusions. Furthermore, publication of this article will hopefully lead to further development of data sets that can be used to conduct further studies in the future. I support the publication of this article in Nature Communications.

I have only a few minor editorial comments for changes to this version:

Line 156: "each variables" should be "each variable"

Line 190: The hyphen in "Forward-" appears to be in error.

Line 246: It's not clear that the word "in" belongs in this line.

Line 372: "the trainee" should read "trainee".

Reviewer #4:

Remarks to the Author:

Through a thorough and detailed revision of the manuscript, claims byLienard and colleagues about success in academia and the role of intellectual synthesis in mentorship now appear to me very solid.

I am satisfied with the pinpoint replies to my two major comments about testing for i) (lack of) trend for successful scientists to go from a graduate mentor to a "better" postdoctoral mentor ii) (no clear) trend for successful scientists who move into a more thriving line of research for the postdoctoral training. The absence of either effect is reported in detail in the main text and in the

supplementary material.

I appreciated that the revised version of the paper maintains all the rigour of the first submission, but it is now much easier to read. This applies in particular to Section 2.2 about model of academic success in life science (mark 8), which is now much more accessible to non statisticians and suitable for the wide readership of Nature Communications.

I also found convincing the discussion of similarity and differences with the paper by Mamgren et al. raised by comments from another reviewer, as well as evidence for the importance of intellectual synthesis both in the neuroscience and other life sciences.

Also, references from the research policy and science of science community have been adequately inserted and discussed in the manuscript (please note that reference 43 arXiv:1607.05606 has recently been published in the Journal of Informetrics).

Taken together, the paper is well-written and timely in its findings, shedding new light on the determinants of academic success (a widely debated topic in the literature) in the life sciences, and I would now recommend acceptance in Nature Communications.

Reviewer #1 (Remarks to the Author):

I commend the authors on all the work they did to improve the manuscript. I think the additional tests dramatically strengthen the confidence on the reported results.

However, I still think that several changes need to be made. First and foremost, I believe that the analyses in the main text should be restricted to neuroscience. The number of triplets for neuroscience is about six-fold the number for all other life science fields.

I think the main text could state that analyses for other life science disciplines are not in conflict with the results for neuroscience but I do believe that the appearance that all results hold for all life science fields is an over-reach.

We agree with this remark, and we made changes to the scope of the paper to more clearly reflect the bias of our dataset toward neuroscience. We altered the text in several places throughout the manuscript, including in the abstract, to be perfectly explicit about it.

In addition, we added more details on our control analysis that compares the neuro and non-neuro parts of the dataset, at mark 3 on page 13 and in Supplementary Materials (mark 4 on page 15). Note that the new analysis considers two data partitions with a more equal size: (a) those who belong exclusively to neuroscience ($n = 14,953$, 62% of the dataset), vs. (b) those who belong to a non-neuro life science field ($n = 5,974$ or 38%, including researchers who categorize themselves in different sub-fields of life science, as long as one of them is not neuroscience). This larger partition demonstrate similar patterns of the data and model coefficients, so it gives more confidence that the intellectual synthesis phenomenon is not restricted to neuroscience.

Overall, even though we gained additional confidence that our results likely generalizes to all of life science, we fundamentally agree with your remark and modified the text to reflect clearly the scope and limits of our analysis.

Second, I believe that the analyses reported in Fig. 3 can be improved. The data in Fig 2A shows that the replication rates of mentors spreads over a range of values that are actually quite different: 0.5 trainees/decade means about two trainees over a 40 year career, 5 trainees/decade means 20 trainees over a 40 year career. But there is even the problem with some mentors likely having even higher replication rates. Instead of plotting the rather uninteresting distributions shown in Figs. 3A-C, I recommend partitioning the mentors into several groups according to their replication rates and showing box plots of values of the different quantities for the different groups (and the results of a test comparing the means for the difference groups, instead of using the KS test). For the case of Fig. 3D, I would plot the value of the mean and skewness versus mentor group.

To the best of our knowledge, nobody has yet reported the distributions of publication similarity, or their link with continuing in academia. We thus believe that the graphs of Figure 3 are an important part of this study, especially as we believe it fits within the flow of the paper. Showing that these distributions are well-behaved also strengthens the confidence in the modeling effort, as it demonstrates that it is not based on pathologically biased/skewed data.

This being said, we understand that the visualization we originally used may look “dry”, so we tried to improve it based on the reviewer’s suggestion, by showing the mean difference and 95% confidence interval (also reinforcing the conclusion of the two-sample Kolmogorov-Smirnov test). We also

investigate specifically the possibility that the beneficial effects of greater/lower publication similarity would only be valid for the high (or low) proliferation mentors. We include this analysis as supplementary information and mention it in main text (mark 1 on page 6).

Third, I realized that I am not sure what the bars and scale represent in Fig. 4 and similar figures. I think it would be more useful to have the bars identifying the 95% CI for the estimates of the coefficients.

We are afraid that an alteration of this figure, introduced in our previous revision, is responsible for the confusion. Indeed, we did not realize that the scale bar (just below the text "scale: z=1") looked like an error bar, suggesting that the error bars represent z-scores (they are really the 95% CI).

We fixed this in all the graphs, in main text and supplementary materials, and rewrote the caption of the graph in main text to avoid any confusion (mark 2 on page 11).

We appreciate, and are thankful for, your reviews which helped us improve our manuscript.

Reviewer #2 (Remarks to the Author):

I am satisfied by the authors' revisions and comments in response to the remarks in my report and those of the other the other two reviewers. While there are limitations to the data set the authors have studied in this work, I believe it is robust enough to support their conclusions. Furthermore, publication of this article will hopefully lead to further development of data sets that can be used to conduct further studies in the future. I support the publication of this article in Nature Communications.

I have only a few minor editorial comments for changes to this version:

Line 156: "each variables" should be "each variable"

Line 190: The hyphen in "Forward-" appears to be in error.

Line 246: It's not clear that the word "in" belongs in this line.

Line 372: "the trainee" should read "trainee".

We corrected the remaining typos.

We kindly thank you for your review and your help in making our manuscript better.

Reviewer #4 (Remarks to the Author):

Through a thorough and detailed revision of the manuscript, claims by Lienard and colleagues about success in academia and the role of intellectual synthesis in mentorship now appear to me very solid.

I am satisfied with the pinpoint replies to my two major comments about testing for i) (lack of) trend for successful scientists to go from a graduate mentor to a "better" postdoctoral mentor ii) (no clear) trend for successful scientists who move into a more thriving line of research for the postdoctoral training. The absence of either effect is reported in detail in the main text and in the supplementary material.

I appreciated that the revised version of the paper maintains all the rigour of the first submission, but it is now much easier to read. This applies in particular to Section 2.2 about model of academic success in life science (mark 8), which is now much more accessible to non statisticians and suitable for the wide readership of Nature Communications.

I also found convincing the discussion of similarity and differences with the paper by Malmgren et al. raised by comments from another reviewer, as well as evidence for the importance of intellectual synthesis both in the neuroscience and other life sciences.

Also, references from the research policy and science of science community have been adequately inserted and discussed in the manuscript (please note that reference 43 arXiv:1607.05606 has recently been published in the Journal of Informetrics).

Taken together, the paper is well-written and timely in its findings, shedding new light on the determinants of academic success (a widely debated topic in the literature) in the life sciences, and I would now recommend acceptance in Nature Communications.

■ *We fixed this reference.*

■ *We appreciate your feedback on our work and we thank you for your review.*

Intellectual Synthesis in Mentorship Determines Success in Academic Careers.

Jean F. Liénard^{1,2*}, Titipat Achakulvisut^{2,3}, Daniel E. Acuna^{3,4} and Stephen V. David¹

¹ Oregon Hearing Research Center, Oregon Health & Science University, Portland, Oregon, United States of America

² Okinawa Institute for Science and Technology, Okinawa, Japan

³ Department of Bioengineering, University of Pennsylvania, Philadelphia, Pennsylvania, United States of America

^{3,4} School of Information Studies, Syracuse University, Syracuse, New York, United States of America

Corresponding author: * jean.f.lienard@gmail.com

Abstract

[revised manuscript text omitted]
 the postdoc training. Arrows-Insets indicate the medians (mean of the differences are significant in all panelstwo trainee groups, $p < 0.001$, i.e., those who do and do not become academic mentors, and 95% confidence intervals). A-B: Trainees who become mentors showed-show greater similarity with their graduate (A) and postgraduate-postdoctoral (B) advisors. C: Lower similarity between mentor these mentors is also linked with better odds to continue in academic research. D: Protégés that have a greater publication proximity with their postdoc mentorsmentor, compared to their graduate mentorsmentor, tend to move more-often to independent academic positions more frequently.

2.3 Model of academic success in life science.

[revised manuscript text omitted]

2.5 Consistency of effects across fields

The composition of the life science dataset is dominated by neuroscience graduates. Indeed, 62% of the
triplets ($n = 14,953$) have a trainee identified as belonging to the field of neuroscience , and the remaining
38% ($n = 5,742$) span several other fields (Fig. 7). To assess the consistency of the effects across fields,
we split the data in two subsets: neuroscience only and other life sciences(Fig. 7). These non-overlapping
datasets show similar, albeit noisier, ~~properties than patterns compared to~~ the full dataset reported in main
text (Fig. 2 vs. Figs. ??-16 and 15 in Supplementary Information). The mentorship patterns of Fig. 2A-D
are comparable across subsets. Both also show the same trends of increasing postdoctoral trainee numbers
and training duration (Fig. 2E-F) and the same patterns of publication similarity (Fig. 2G-H). Importantly,
models computed for both data subsets showed the positive effect of intellectual synthesis, with a strong
effect of mentor publication similarity in both cases (Figs. 18 and 17 in Supplementary Information). Thus,
the advantage of trainees performing intellectual synthesis generalizes across the life sciences.~~We do observe~~
~~some discrepancies between the data subsets. Fewer variables achieve significance in the non-neuroscience~~
~~subset, particularly for the long-term prediction of trainee proliferation rate, possibly reflecting ,
[revised manuscript text omitted]

2. Akil, H., Balice-Gordon, R., Cardozo, D. L., Koroshetz, W., Norris, S. M. P., Sherer, T., Sherman, S. M. and Thiels, E. 2016, ‘Neuroscience training for the 21st century’, *Neuron* **90**(5), 917–926.
3. Aumann, R. J. 1989, Game theory, *in* ‘Game Theory’, Springer, pp. 1–53.
4. Austin, J. 2013, ‘Want to be a professor? choose math’.
5. Barres, B. A. 2013, ‘How to Pick a Graduate Advisor’, *Neuron* **80**(2), 275 – 279.
6. Cameron, A. C. and Trivedi, P. K. 2013, *Regression analysis of count data*, Vol. 53, Cambridge university press.
7. Chariker, J. H., Zhang, Y., Pani, J. R. and Rouchka, E. C. 2016, ‘Identification of Successful Mentoring Communities using Network-based Analysis of Mentor-Mentee Relationships across Nobel Laureates’, *bioRxiv* p. 075432.
8. Check, H. E. 2016, ‘Young scientists ditch postdocs for biotech start-ups.’, *Nature* **539**(7627), 14.
9. Chen, J., Kim, M. and Liu, Q. 2016, ‘Do Female Professors Survive the 19th-Century Tenure System?: Evidence from the Economics Ph. D. Class of 2008’.
10. Chevan, A. and Sutherland, M. 1991, ‘Hierarchical partitioning’, *The American Statistician* **45**(2), 90–96.
11. Cleveland, W. S., Grosse, E. and Shyu, W. M. 1992, ‘Local regression models’, *Statistical models in S* **2**, 309–376.
12. Cohen, S., Dror, G. and Ruppin, E. 2007, ‘Feature selection via coalitional game theory’, *Neural Computation* **19**(7), 1939–1961.
13. Crane, D. 1965, ‘Scientists at major and minor universities: A study of productivity and recognition’, *American sociological review* pp. 699–714.
14. Crosta, P. M. and Packman, I. G. 2005, ‘Faculty productivity in supervising doctoral students’ dissertations at Cornell University’, *Economics of Education Review* **24**(1), 55–65.
15. David, S. V. and Hayden, B. Y. 2012, ‘Neurotree: A collaborative, graphical database of the academic genealogy of neuroscience’, *PLoS one* **7**(10), e46608.
16. Deerwester, S., Dumais, S. T., Furnas, G. W., Landauer, T. K. and Harshman, R. 1990, ‘Indexing by latent semantic analysis’, *Journal of the American society for information science* **41**(6), 391.
17. Dutt, K., Pfaff, D. L., Bernstein, A. F., Dillard, J. S. and Block, C. J. 2016, ‘Gender differences in recommendation letters for postdoctoral fellowships in geoscience’, *Nature Geoscience* **9**(11), 805–808.

- 18. Efron, B. 1979, 'Bootstrap methods: another look at the jackknife', *The annals of Statistics* pp. 1–26.
- 19. Fortunato, S., Bergstrom, C. T., Börner, K., Evans, J. A., Helbing, D., Milojević, S., Petersen, A. M.,
Radicchi, F., Sinatra, R., Uzzi, B. et al. 2018, 'Science of science', *Science* **359**(6379), eaaa0185.
- 20. Freeman, R. B. 2002, 'Thanks for the great postdoc bargain', *Science's Next Wave* .
- 21. Gelman, A., Hwang, J. and Vehtari, A. 2014, 'Understanding predictive information criteria for Bayesian
models', *Statistics and Computing* **24**(6), 997–1016.
- 22. Ginther, D. K. and Kahn, S. 2006, 'Women's careers in academic social science: Progress, pitfalls, and
plateaus', *The Economics of Economists, A. Lanteri and J. Vromen, eds.(Cambridge, UK: Cambridge*
*University Press, 2014)* .
- 23. Granovetter, M. 1995, *Getting a job: A study of contacts and careers*, University of Chicago Press.
- 24. Granovetter, M. S. 1973, 'The strength of weak ties', *American journal of sociology* **78**(6), 1360–1380.
- 25. Hall, R. M. and Sandler, B. R. 1983, 'Academic mentoring for women students and faculty: A new look
at an old way to get ahead.'
- 26. Hofmann, T. 1999, Probabilistic latent semantic indexing, *in* 'Proceedings of the 22nd annual interna-
tional ACM SIGIR conference on Research and development in information retrieval', ACM, pp. 50–57.
- 27. Kahn, S. and Ginther, D. K. 2017, 'The impact of postdoctoral training on early careers in biomedicine',
*Nature Biotechnology* **35**(1), 90–94.
- 28. Kram, K. E. 1988, *Mentoring at work: Developmental relationships in organizational life.*, University
Press of America.
- 29. Kruskal, W. 1987, 'Relative importance by averaging over orderings', *The American Statistician* **41**(1), 6–
10.
- 30. Lan, X. 2012, 'Permanent visas and temporary jobs: evidence from postdoctoral participation of foreign
PhDs in the United States', *Journal of Policy Analysis and Management* **31**(3), 623–640.
- 31. Lane, J. and Bertuzzi, S. n.d., The star metrics project: current and future uses for s&e workforce data.
- 32. Layton, R. L., Brandt, P. D., Freeman, A. M., Harrell, J. R., Hall, J. D. and Sinche, M. 2016, 'Diversity
exiting the academy: Influential factors for the career choice of well-represented and underrepresented
minority scientists', *CBE-Life Sciences Education* **15**(3), ar41.
- 33. Long, J. and McGinnis, R. 1985, 'The effects of the mentor on the academic career', *Scientometrics*
**7**(3-6), 255–280.
- 34. Long, J. S., Allison, P. D. and McGinnis, R. 1979, 'Entrance into the academic career', *American*
*sociological review* pp. 816–830.
- 35. Malmgren, R. D., Ottino, J. M. and Amaral, L. A. N. 2010, 'The role of mentorship in protégé perfor-
mance', *Nature* **465**(7298), 622–626.

- 36. Martinez, E. D., Botos, J., Dohoney, K. M., Geiman, T. M., Kolla, S. S., Olivera, A., Qiu, Y., Rayasam,
G. V., Stavreva, D. A. and Cohen-Fix, O. 2007, ‘Falling off the academic bandwagon’, *EMBO reports*
**8**(11), 977–981.
- 37. NAS 2014, ‘National Academy of Sciences, National Academy of Engineering, Institute of Medicine. The
Postdoctoral Experience Revisited’, *National Academies Press* .
- 38. Nerad, M. and Cerny, J. 1999, ‘Postdoctoral Patterns, Career Advancement, and Problems’, *Science*
**285**(5433), 1533–1535.
- 39. Nihm, S. D. 1976, ‘Polynomial law of sensation.’, *American Psychologist* **31**(11), 808.
- 40. NRC 1981, *National Research Council (United States), Committee on a Study of Postdoctorals in Science*
*and Engineering in the United States and National Research Council (US), Commission on Human*
*Resources. Postdoctoral Appointments and Disappointments: A Report of the Committee on a Study of*
*Postdoctorals in Science and Engineering in the United States*, number 3132, National Academy Press.
- 41. NSB 2016, ‘National Science Board: Science and engineering indicators 2016’, *NS Foundation (Ed.)*.
*Arlington, VA: National Science Foundation* .
- 42. Pan, R. K., Petersen, A. M., Pammolli, F. and Fortunato, S. ~~2016~~2018, ‘The memory of science: In-
flation, myopia, and the knowledge network’, ~~*arXiv preprint arXiv:1607.05606*~~*Journal of Informetrics*
12(3), 656–678.
- 43. Pedregosa, F., Varoquaux, G., Gramfort, A., Michel, V., Thirion, B., Grisel, O., Blondel, M., Pretten-
hofer, P., Weiss, R., Dubourg, V. et al. 2011, ‘Scikit-learn: Machine learning in Python’, *Journal of*
*Machine Learning Research* **12**(Oct), 2825–2830.
- 44. Petersen, A. M., Fortunato, S., Pan, R. K., Kaski, K., Penner, O., Rungi, A., Riccaboni, M., Stanley,
H. E. and Pammolli, F. 2014, ‘Reputation and impact in academic careers’, *Proceedings of the National*
*Academy of Sciences* **111**(43), 15316–15321.
- 45. Petersen, A. M., Riccaboni, M., Stanley, H. E. and Pammolli, F. 2012, ‘Persistence and uncertainty in
the academic career’, *Proceedings of the National Academy of Sciences* **109**(14), 5213–5218.
- 46. Powell, K. 2015, ‘The future of the postdoc’, *Nature* **520**(7546), 144.
- 47. Ragins, B. R., Kram, K. E., Ragins, B. and Kram, K. 2007, ‘The roots and meaning of mentoring’, *The*
*handbook of mentoring at work: Theory, research, and practice* pp. 3–15.
- 48. Rogers, E. M. 2010, Diffusion of innovations, Simon and Schuster, chapter 8.
- 49. Shapley, L. 1953, ‘A value for n-person games’, *Contributions to the Theory of Games (Edited by H. W.*
*Kuhn and A. W. Tuck* **2**, 307–317.
- 50. Sinatra, R., Wang, D., Deville, P., Song, C. and Barabási, A.-L. 2016, ‘Quantifying the evolution of
individual scientific impact’, *Science* **354**(6312), aaf5239.

- 51. Smyth, P. 2000, ‘Model selection for probabilistic clustering using cross-validated likelihood’, *Statistics*
*and computing* **10**(1), 63–72.
- 52. Steinpreis, R. E., Anders, K. A. and Ritzke, D. 1999, ‘The impact of gender on the review of the curricula
vitae of job applicants and tenure candidates: A national empirical study’, *Sex roles* **41**(7), 509–528.
- 53. Stone, M. 1977, ‘An asymptotic equivalence of choice of model by cross-validation and Akaike’s criterion’,
*Journal of the Royal Statistical Society. Series B (Methodological)* pp. 44–47.
- 54. Stufken, J. 1992, ‘Letters to the Editor: On Hierarchical Partitioning’, *The American Statistician*
**46**(1), 70–77.
- 55. Su, X. 2013, ‘The impacts of postdoctoral training on scientists’ academic employment’, *The Journal of*
*Higher Education* **84**(2), 239–265.
- 56. Sugimoto, C. R. and Cronin, B. 2012, ‘Biobibliometric profiling: An examination of multifaceted
approaches to scholarship’, *Journal of the American Society for Information Science and Technology*
**63**(3), 450–468.
- 57. Trix, F. and Psenka, C. 2003, ‘Exploring the color of glass: Letters of recommendation for female and
male medical faculty’, *Discourse & Society* **14**(2), 191–220.
- 58. Waltman, L. 2016, ‘A review of the literature on citation impact indicators’, *Journal of Informetrics*
**10**(2), 365–391.
- 59. Way, S. F., Morgan, A. C., Clauset, A. and Larremore, D. B. 2017, ‘The misleading narrative of the
canonical faculty productivity trajectory’, *Proceedings of the National Academy of Sciences* p. 201702121.
- 60. Yang, L. and Webber, K. L. 2015, ‘A decade beyond the doctorate: the influence of a US postdoctoral
appointment on faculty career, productivity, and salary’, *Higher Education* **70**(4), 667–687.
- 61. Yin, X., Han, J. and Philip, S. Y. 2007, Object distinction: Distinguishing objects with identical names,
in ‘Data Engineering, 2007. ICDE 2007. IEEE 23rd International Conference on’, IEEE, pp. 1242–1246.
- 62. Zeileis, A., Kleiber, C. and Jackman, S. 2008, ‘Regression models for count data in R’, *Journal of*
*statistical software* **27**(8), 1–25.

Supplementary information

1

2 Availability of date information

Availability of date information	
Start of PhD	51%
End of PhD	66%
Earliest publication date with graduate advisor	74%
At least one graduate date available	90%
Start of Postdoc	57%
End of Postdoc	39%
Earliest publication date with graduate advisor	71%
At least one postdoc date available	89%

Table 2: Statistics of data availability based on publications and manually entered dates

3 **6** Test for interaction between mentor proliferation rate and publication
4 similarity

Figure 8: Difference in publication similarity between the group of trainees who also mentored at least one trainee in turn, and the group of trainees without mentoring reported in the dataset. The metrics shown are the same as in Fig. 3 in main text, points represent the average difference and the error bars the 95% confidence intervals. “Full dataset” refers here to all mentors. “Lower half” and “Upper half” show these differences when computed only on the low-proliferation and high-proliferation mentors (split based on the highest proliferation rate of the pair). Overall, we see similar patterns as the ones shown in Fig. 3, i.e. a beneficial pattern of higher similarity of trainees with individual mentors on continuation (red and blue traces), and detrimental patterns associated with mentor publication similarity (green) and higher proximity with graduate advisor compared to postdoctoral advisor (purple). There is no clear connection between the publication similarity patterns and mentor proliferation rates.

7 Model and variable selection

Two model composition techniques are suitable to handle the large prevalence of postdocs without trainees, ~~the~~ the hurdle and *zero-inflated* frameworks⁶. ~~They differ by their modeling of~~ These models differ in how they account for researchers without trainees: ~~the~~. The hurdle framework assumes that all independent researchers have at least one trainee, while the zero-inflated framework allows the existence of some independent researchers that have no trainee in the database. This latter scenario would correspond either to incomplete Academic Tree profiles or to researchers not involved in graduate/postdoctoral training. Furthermore, the proliferation rate may be modeled as a Poisson distribution or as a negative binomial distribution. The former assumes that count variance is directly proportional to mean count, while the latter relaxes this assumption and allows over-dispersion, at the cost of an extra free parameter.

To decide which of the four model architectures (hurdle or zero-inflated with Poisson or negative binomial distribution) and which predictors yielded the best fit to the data, we screened the performance of each architecture on all possible combinations of predictors. This was done by calculating parameter values for each model that maximized log-likelihood of predicted outcomes, using a 10-fold cross-validation scheme. Shapley values were then computed scoring the relative contribution of each variable to the overall model's performance. Negative binomial distributed count rates consistently outperformed count rates conforming to a Poisson distribution, indicating the presence of over-dispersion in rates for researchers with the highest proliferation rate. Zero-inflated models performed better than hurdle models, indicating that an assumption that all independent research faculty have at least one trainee is not consistent with this dataset (Table 3). The best-fitting model overall was the zero-inflated negative binomial model. Cross-validated predictions for each input variable are shown in Fig. 19 in Supplementary Information. Given its superior performance, we focus on this mathematical model for the main results of the paper.

$\Delta\mathcal{L}$ in cross-validation				
	HP	ZIP	HNB	ZINB
temporal	-10.24	-10.23	-0.47	-0.45
network	-9.45	-9.40	-0.54	-0.30
publications	-10.42	-10.38	-0.71	-0.52
temporal + network	-9.12	-9.11	-0.07	-0.05
temporal + publications	-10.10	-10.09	-0.39	-0.38
network + publications	-9.31	-9.26	-0.47	-0.24
temporal + network + publications	-8.97	-8.96	-0.03	0.00

Table 3: Impact of model ~~types~~ architecture and ~~predictors~~ parameter type on ~~predictive~~ prediction accuracy. The ~~mathematical~~ models compared are: ~~the~~ hurdle Poisson ~~model~~ (HP), hurdle negative binomial ~~models~~ (HNB), zero-inflated Poisson ~~models~~ (ZIP) and zero-inflated negative binomial ~~models~~ (ZINB). Predictors were grouped ~~under~~ into the same categories as in Table 1: “temporal” for the year of the end of the postdoctoral appointment as well as the total duration of postdoctoral training; “network” for the proliferation ~~rate~~ of mentors, their academic age at training and their common ancestor distance; and “publications” for the ~~publication~~ semantic similarity between mentors’ publications (prior to meeting the trainee) and ~~the~~ publication ~~similarity~~ between those of the postdoctoral trainee and mentor (prior to training). The values displayed are cross-validated ~~log-likelihood~~ log-likelihoods aligned on the best model (“ZINB” model using “network + publications + temporal” variables), with higher values denoting more accurate models.

8 Influence of graduate mentor ~~'s academic age~~only

Fig. 9 show the regression coefficients computed using only features from graduate studies (Fig. 9). We
 find similar coefficients in these partial regressions, showing that the effects of graduate mentor features
 are independent from the effects of postdoc mentor features. Of interest, graduate mentor age is still non-
 significant in this analysis, showing that ~~the lack of a possible~~ impact of graduate mentor age is not ~~caused~~
 ~~by masked by correlated~~ features from the postdoctoral mentor. ~~As this extra~~ This additional analysis was
 computed on the same triplet dataset ~~, used in the rest of this study, and~~ it does not ~~rule out the more~~
 ~~general potential impact of features from the graduate advisor~~ include graduate mentor / trainee dyad, in
 ~~encouraging/discouraging a PhD to continue to the postdoc step~~ dyads without a postdoctoral mentor. There
 ~~may be different effects of graduate mentors in this dyad group, as graduate students can move directly to~~
 ~~independent positions or choose to discontinue training before pursuing a postdoctoral fellowship.~~

Figure 9: Coefficients of a model focused on graduate mentor features (academic age, training rate, co-publishing and similarity with trainee) and excluding features related to postdoctoral studies.

9 Alternative models of postdoctoral **preponderance**training effects

Figure 10: Alternative model computed with an additional interaction term, “Postdoc ÷ graduate mentor rates”, which was designed to be high for trainees that moved to a ~~more prolific~~ postdoctoral mentor with higher proliferation rate (“upward mobility” hypothesis). The lack of significance ~~of for~~ this additional term showed that there is no systematic benefit associated with such a strategy.

Figure 11: Alternative model computed with a new interaction term, “Postdoc mentor rate × similarity”, contrasted with a second term: “Graduate mentor rate × similarity”. The first term should be high for trainees who moved into the “better” subfield of their postdoc mentor, while the second term contrasts this effect and should be high for trainees who stayed in the “better” subfield of their graduate studies. These terms were not linked to increased (or decreased) odds of finding a permanent position in this dataset. Interestingly, there is a slight influence of the “Postdoc mentor rate × similarity” term on long-term proliferation. One interpretation is that it corresponds to a long-term fatigue effect of disengagement from opportunistic trainees who embraced the research line of their postdoctoral mentor. However, it may also be a spurious effect that parallels the increased long-term effect of the “Postdoc mentor rate” in this regression, compared to the original regression.

10 Time-dependence of regression coefficients

Fig. 12 shows the regression coefficients obtained when training the model on temporal subsets of the data
and without the time-controlling variable ~~of~~ “postdoc end year”. Except for this omission, the variables
included in the model were the ones obtained after the main selection process (cf. Table 1 in main text).
The optimized coefficients from Fig. 12 display much more variability than with the regression shown in main
text, due to lower sample sizes and the exclusion of temporal variables, but overall display similar trends as
the full model.

We also controlled for long term temporal effects through a series of higher-order terms into the regression
model (Fig. 13). This expansion of higher-order terms has the advantage of modeling arbitrary temporal
trends, as in Taylor series, to the risk of over-fitting temporal trends³⁹. ~~Here, not surprisingly~~ Not surprisingly,
these additional terms were able to capture some additional variance in the data, but they had no large impact
on the other factors of interest in the regression. In particular, the intellectual synthesis effects appeared
robust to the inclusions of finer temporal controls.

Figure 12: Regression coefficients obtained when optimizing the model on different temporal subsets while removing the postdoc end date from the set of predictors. Temporal partitions were chosen to contain roughly equal sample sizes, and were as follows: 1980 – 1992 ($n = 1,436$); 1992 – 1998 ($n = 1,337$); 1998 – 2002 ($n = 1,347$); 2002 – 2005 ($n = 1,404$); 2005 – 2007 ($n = 1,343$). The coefficients estimated to predict continuation in academia (A) and long-term mentoring rate (B) are consistent with the ones obtained with the full model that includes temporal predictors (represented in these plots as shaded areas). Specifically, whenever a variable reaches significance within a temporal subset, it does so in the same direction as the trend apparent in the full model. Reciprocally, whenever a variable reaches significance in the full model, there is at least one temporal subset when it reaches significance as well.

Figure 13: Model coefficients, similar to Fig.4 in main text but with the inclusion of more complex temporal trends, captured here through a series of ~~higher-order~~ higher order polynomial terms: (Training end year)², (Training end year)³ and (Training end year)⁴.

We also tried another ~~type of non-linear modeling of~~ model for nonlinear temporal effects, by replacing
the continuous time variable (“~~Training training~~ end year”) ~~by with~~ an ordinal variable capturing the **same**
~~temporal subdivisions as~~ temporal subdivisions in Fig.4 (“~~Training training~~ end epoch”, numbered from 1
to 5). This transformation did not affect which coefficients were significant, nor did it change the order of
magnitude of their effects (Fig. 14).

Figure 14: Model coefficients, similar to Fig.4 in main text but with a simpler temporal modeling that uses the ordinal variable “~~Training~~ training end epoch”.

**11 Consistency of effects across fields**

4 ↑

Figure 15: Same-Main features of mentorship triplets, plotted as in Fig. 2 in the main text, but using only data from non-neuroscience for neuroscience graduates ($n=3,271$ - $n=14,953$ triplets).

Figure 16: Same Main features of mentorship triplets, plotted as in Fig. 2 in the main text, but using data the general population of life science trainees (complementary set of the one analyzed in Supplementary Fig. 15). This includes multi-disciplinary life science researchers, as long as at least one reported field is not neuroscience graduates ($n = 14,953$ - $n = 5,742$ triplets).

Figure 17: Same as Fig. 4 in main text, but using data of neuroscience graduates (corresponding to Supplementary Fig. 15).

Figure 18: Same as Fig. 4 in main text, but using data from general life science graduates, including multi-disciplinary researchers (corresponding to Supplementary Fig. 16). Note that as this dataset also includes multi-disciplinary researchers belonging to different fields including neuroscience, we gave a half weight to this data in the regression to allow comparison with the “neuroscience only” dataset in Supplementary Fig. 17.

12 Cross-validated model predictions and data

Figure 19: Academic continuation in life science Detailed prediction analysis for probability of obtaining an independent research position (panels A to F) and modeled training-proliferation rate (panels G to I), explained by the most significant variables. For each variable, the data is shown in black, and the cross-validated predictions prediction of the model, including this variable, is shown in orange (“full model”). To visualize the contribution of each variable, we also display the cross-validated predictions of the model without this variable in purple (“partial model”). Lines and shaded areas represent respectively the mean values and their 95% bootstrapped confidence interval.

Figure 20: Contribution of post-training publication similarity variables to the odds of continuing an academic career. ~~In each of these plots, the data is shown in black, for graduate (A) and the cross-validated predictions of the model including this variable is shown in orange postdoctoral mentors (“full model” B). To visualize the contribution of each variable, we also display the cross-validated predictions of the model without this variable plotted as in purple (“partial model”). Lines and shaded areas represent respectively the mean values and their 95bootstrapped confidence intervals supplemental Fig. Overall, proximity with the postdoctoral advisor has a larger influence than proximity with the graduate advisor 19. Also, the mentorsMentor/trainee proximity similarity displays a larger explanatory power stronger effect when computed on publication data is included from after the end of training (compared to using only data available at the end of the postdoc, cf. supplemental -Fig. 19F). Also, with higher similarity linked to better odds the postdoctoral mentor continues to have a larger influence than that of continuing in research the graduate advisor. —~~

Reviewers' Comments:

Reviewer #1:

Remarks to the Author:

I believe that the authors have thoughtfully address all of my prior concerns. I think this is an important study that deserves publication in Nature Communications.